# Cloud Detection over Snow and Ice with Oxygen A- and B-band Observations from the Earth Polychromatic Imaging Camera (EPIC)

Yaping Zhou[1,2], Yuekui Yang[1], Meng Gao[1,3], Peng-Wang Zhai[4]

[1]NASA Goddard Space Flight Center, Greenbelt, MD.

[2]JCET/University of Maryland Baltimore County, Baltimore, MD
[3]SSAI/NASA Goddard Space Flight Center, Ocean Ecology Laboratory,

Greenbelt, MD, USA

[4]JCET/Department of Physics, University of Maryland Baltimore County,

Baltimore, MD, 21250, USA

*Correspondence to*: Yaping Zhou (yaping.zhou-1@nasa.gov)

## Abstract

Satellite cloud detection over snow and ice has been difficult for passive remote sensing instruments due to the lack of contrast between clouds and cold/bright surfaces; cloud mask algorithms often heavily rely on shortwave IR channels over such surfaces. The Earth Polychromatic Imaging Camera (EPIC) onboard the Deep Space Climate Observatory (DSCOVR) does not have infrared channels, which makes cloud detection over snow and ice surfaces even more challenging. This study investigates the methodology of applying EPIC's two oxygen absorption band pair ratios in A-band (764 nm, 780 nm) and B-band (688 nm, 680 nm) for cloud detection over the snow and ice surfaces. We develop a novel elevation and zenith angle-dependent threshold scheme based on radiative transfer model simulations that achieves significant improvements over the existing algorithm. When compared against a composite cloud mask based on geosynchronous Earth orbit (GEO) and low Earth orbit (LEO) sensors, the positive detection rate over snow and ice surfaces increased from around 36% to 65% while the false detection rate dropped from 50% to 10% for observations of January 2016 and 2017. The improvement in July is less substantial due to relatively better performance in the current

algorithm. The new algorithm is applicable for all snow and ice surfaces including Antarctic, sea ice, high-latitude snow, and high-altitude glacier regions. This method is less reliable when clouds are optically thin or below 3 km because the sensitivity is low in oxygen band ratios for these cases.

## 1. Introduction

The Earth Polychromatic Imaging Camera (EPIC) onboard the Deep Space Climate Observatory (DSCOVR) was launched in 2015. The unique orbit of DSCOVR allows the EPIC instrument to take continuous measurements of the entire sunlit side of the Earth from the nearly backscattering direction (scattering angles between 168.5° and 175.5°) from the first Lagrangian (L1) point of the Earth-Sun orbit, approximately 1.5 million km away. The EPIC instrument has 10 narrow spectral channels in the ultra-violet (UV) and visible/near-infrared (Vis/NIR) (317-780 nm) spectral range that enable retrieval of atmospheric ozone, cloud, and surface vegetation information. The focal plane of the EPIC system is a 2048 × 2048 pixel charge-coupled device (CCD) array that covers the entire disk with a nadir resolution of 8 km. However, due to the limited transmission capacity, all channels except the 443 nm channel are reduced to 1024 x 1024 arrays through onboard processing and interpolated back to full resolution after being downlinked. The operation of the instrument and the downlink speed limit the temporal frequency of measurements to be approximately once every 1.5 and 2.5 hours in boreal winter and summer, respectively. Detailed descriptions of the EPIC instrument can be found in Herman et al. (2018), Marshak et al. (2018), and Yang et al. (2019).

The EPIC cloud product, including cloud mask (CM), cloud effective pressure (CEP), cloud effective height (CEH), and cloud optical thickness (COT), are developed with fewer spectral channels compared with many spectroradiometers currently onboard the polar and geostationary satellites (Yang et al., 2019). For example, the Moderate-resolution Imaging Spectroradiometer (MODIS) cloud algorithm uses simultaneous two-channel retrievals of COT and cloud effective radius (CER) separately for water and ice clouds, with the cloud phase pre-determined by more spectral tests. Since EPIC does not have a particle size-sensitive channel, and has limited capability to determine the cloud phase, the EPIC COT retrieval uses a single channel and

derives two sets of COT, one for assumed ice phase and one for assumed liquid phase, each with fixed CER (Yang et al., 2019; Meyer et al., 2016). CEP is derived based on two oxygen ($O_2$) band pairs, each consisting of an absorption and a reference channel. The A-band absorption channel is centered at 764 nm with a Full Width at Half Maximum (FWHM) of 1.02 nm, and its

reference channel is centered at 780 nm with a FWHM of 1.8 nm. The B-band's absorption channel is centered at 688 nm with a FWHM of 0.84 nm, and its reference channel is centered at 680 nm with a FWHM of 1.6 nm (Marshak et al. 2018). The $O_2$ absorption bands are sensitive to cloud height because the presence of clouds, especially thick clouds, reduces the absorbing air mass that light travels through; hence, the ratio of the absorbing and reference bidirectional

reflectance functions (BRF) becomes larger. Since $O_2$ absorption at 764 nm is stronger than at 688 nm, the A-band ratio has higher sensitivity than the B-band ratio (Yang et al., 2013).

Satellite cloud detections are usually based on the contrast between clouds and the underlying earth surface. Clouds are generally higher in reflectance and lower in temperature

than the surface, which makes simple threshold approaches in the visible and infrared window channels effective in cloud detection (e.g., Saunders and Kriebel, 1988; Rossow and Garder, 1993; Yang et al., 2007; Ackerman et al., 2010). However, there are many situations when simple visible and infrared threshold tests are not able to separate clouds from surface or from heavy atmospheric aerosols such as dust and smoke. The contrasts between clouds and

surface are weak in the visible channels when the surface is bright, and weak in the IR channels when the surface temperature is very low or the cloud is very low in height. Additionally, partially cloudy pixels due to small-scale cumulus or cloud edges also increase the detection difficulty. The official MODIS CM algorithm uses more than 20 spectral channels to detect clouds in various situations. In particular, it heavily relies on shortwave infrared channels at

1.38, 1.6, 2.1μm and thermal channels at 11 and 13.6 μm for cloud detection over snow and ice (Frey et al., 2008; Ackerman et al., 2010)

The lack of infrared and near-infrared channels in EPIC makes cloud detection very challenging, especially over snow and ice surfaces. The current EPIC CM algorithm adopts a

general threshold method, which uses two sets of spectral tests for each of the three scene types: ocean, land, and ice/snow (Yang et al., 2019). Over ocean, the 680 nm and 780 nm

channels are used for cloud detection, because clouds and the sea surface contrast well in both channels. Over land, because of large variations in surface reflectivity at 680 nm and 780 nm, these two channels can no longer be used alone for cloud detection. Instead, the algorithm uses the 388 nm channel and the A-band reflectivity ratio, i.e., $R_{764}/R_{780}$ for cloud detection. The 388 nm channel is used because of its low reflectivity over land surfaces. The A-band ratio is used based on the same mechanism as the cloud height retrieval because clouds reduce $O_2$ band absorption by increasing the height of the effective reflective layer. Thus, the A-band ratio of a cloudy pixel is expected to be higher than that of a clear pixel in an otherwise identical situation. The A-band ratio is selected for use over the land surface because it has higher sensitivity than the B-band ratio. Over snow- and ice-covered regions, the $O_2$ A- and B-band ratios are used for cloud detection since the contrast between surface and clouds is small in the visible and UV channels. Evaluation using the collocated cloud retrievals from other sensors show that the EPIC CM performs very well in general. The EPIC CM has an overall 80.2% accuracy rate and 85.7% correct cloud detection rate (accuracy and correct cloud detection rate are defined in Section 5), but a large discrepancy is found over the snow- or ice-covered surfaces where the EPIC algorithm significantly underestimates cloud fraction, especially over the ice and snow-covered Antarctica (Yang et al., 2019). One of the reasons is that the current algorithm uses empirically derived fixed A-band and B-band ratio thresholds without considering the photon path changes due to sun/sensor geometry and surface elevation.

The current work aims to improve EPIC cloud masking through a better understanding of the variability of the $O_2$ band ratios under various clear and cloudy conditions over snow and ice surfaces. Radiative transfer model simulations and observed reflectance will be examined to derive dynamic thresholds for the $O_2$ band ratios so that the new algorithm is applicable to all snow and ice surfaces, i.e., Antarctica, Greenland, snow in high latitude and glaciers over high mountains.

To compute radiation fluxes from EPIC and NISTAR instruments on board the DSCOVR satellite (Su et al. 2018, 2019), the Clouds and the Earth's Radiant Energy System (CERES) team at NASA Langley Research Center created a composite cloud product from GEO/LEO satellites by projecting the GEO/LEO retrievals to the EPIC grid at each EPIC observing time (Khlopenkov et al., 2017). The procedure ensures that every EPIC image/pixel has a corresponding GEO/LEO composite image/pixel with approximately same size and observation

time. The LEO satellites include NASA Terra and Aqua MODIS and NOAA AVHRR while geosynchronous satellite imagers include the Geostationary Operational Environmental Satellites (GOES) operated by NOAA, Meteosat satellites by EUMETSAT, and Multifunctional Transport Satellites (MTSAT) and Himawari-8 satellites operated by the Japan Meteorological Agency

(JMA). Compared to EPIC, the GEO/LEO sensors are usually better equipped for cloud detection over snow and ice. For this study, the GEO/LEO cloud mask is used as a reference for EPIC threshold finding and result comparison purposes. The time differences between the GEO/LEO and the EPIC observations are included in the product files. To limit uncertainties, we only use pixels where the GEO/LEO and EPIC observations are within 5 minutes of each other.

The remainder of the paper is organized as follows: Section 2 provides an analytical discussion on the relationship between the $O_2$ band ratios with the relative airmass and surface elevation. Section 3 conducts sensitivity studies through radiative transfer modeling, and describes the threshold derivation procedure using the model simulations. Section 4 describes the

new cloud mask algorithm for the EPIC instrument over snow and ice. Section 5 reports on the new algorithm validation. Finally, Section 6 provides a brief summary and discussion.

2.  An analytical guide with monochromatic radiative transfer

Oxygen absorption has been applied to remote sensing of cloud and aerosol extensively (e.g., Grechko, et al. 1973; Fischer, J. and Grassl, 1991; Min et al. 2004; Stammes et al., 2008; Wang et al., 2008; Vasilkov et al. 2008; Ferlay et al., 2010; Yang et al. 2013; Ding et al. 2016; Richardson et al, 2019). The underlying physics is based on the well-known gaseous absorption of well-mixed atmospheric $O_2$. Changes in observed radiance in the $O_2$ band are expected to

contain information on how clouds or atmospheric aerosols interrupt the normal absorption photon path and/or provide additional scattering at different vertical levels. The cloud detection using the $O_2$ absorption band ratios is based on the fact that clouds decrease the photon path length within the atmosphere. Clouds reduce the oxygen absorption optical thickness while their impact on the nearby reference channels is negligible. As a result, holding everything else equal,

the BRF ratios between the absorption and the reference channels are expected to be larger for cloudy skies than clear skies. In reality, photon paths can be very complicated: Yang et al. (2013)

listed six pathways for a photon to reach the sensor. To simplify the discussion, we focus only on completely clear or cloudy cases. To determine a threshold for separating clear sky and cloudy sky, the first step is to understand factors that affect the clear sky $O_2$ band ratios. The second step is to understand how $O_2$ band ratios change with the presence of different kinds of clouds. This step helps determine where thresholds can be drawn between clear skies and cloudy skies, and what kind of sensitivity or uncertainty can be expected with this method.

The radiances entering the sensor consist of many components, including sunlight directly reflected by clouds, aerosols, and surfaces, as well as Rayleigh scattering through single- and multiple-scattering processes. Rayleigh optical thickness at the Oxygen A- and B-band regions are about 0.02 and 0.04, respectively. Hence, for a clear sky over a bright surface, we can neglect the contribution of single and multiple scattering. Thus, the monochromatic BRF at the top of atmosphere can be related to the column optical depth via Beer's Law as:

$$R_{abs} = T_{abs}^{dn} * \alpha_{abs} * T_{abs}^{up} = \alpha_{abs} e^{-\left(\tau(z) + \tau_{ray}(z)\right)\left(\frac{1}{\mu} + \frac{1}{\mu_0}\right)} \tag{1}$$

$$R_{ref} = T_{ref}^{dn} * \alpha_{ref} * T_{ref}^{up} = \alpha_{ref} e^{-\tau_{ray}(z)\left(\frac{1}{\mu} + \frac{1}{\mu_0}\right)} \tag{2}$$

$$m = \frac{1}{\mu} + \frac{1}{\mu_0} = \frac{1}{\cos\theta} + \frac{1}{\cos\theta_0} \tag{3}$$

where $R_{abs}$ and $R_{ref}$ are the BRF for the oxygen band and its reference band, respectively. BRF at the top of the atmosphere is a product of downward transmittance ($T_{dn}$), spectral surface reflection albedo $\alpha$, and upward transmittance ($T_{up}$). $\tau$ and $\tau_{ray}$ are optical thickness values due to $O_2$ absorption and Rayleigh scattering at nadir, respectively, and are functions of surface elevation $Z$. $m$ is the total airmass accounting for the slant path for both incoming ($T_{dn}$) and reflected light ($T_{up}$). The absorption channels are subject to both absorption and Rayleigh scattering, while the reference channels only incur Rayleigh scattering. The ratio of $R_{abs}$ and $R_{ref}$ led to cancellation of Rayleigh scattering and surface albedo since the two channels are very close, such that

$$\frac{R_{\mathrm{abs}}}{R_{\mathrm{ref}}} = e^{-\tau(z)\left(\frac{1}{\mu}+\frac{1}{\mu_0}\right)} = e^{-\tau(z)*m} \qquad (4)$$

The absorption optical thickness at a given location decreases exponentially with surface elevation following the approximate relationship in Eq. (5) (Petty, 2006):

$$\tau(z) = K_a w_1 \rho_0 H \exp\left(-\frac{Z}{H}\right) = c * \exp\left(-\frac{Z}{H}\right) \qquad (5)$$

Here $H$ is the scale height, and $K_a, w_1, \rho_0$ are the mass absorption coefficient, mixing ratio of oxygen, and density of air at sea level, respectively. $c = K_a w_1 \rho_0 H$, and can be assumed constant

10   for our problem. To relate the O2 band ratios directly to surface elevation and zenith angles in two separate terms, we take a double logarithm on both sides of Eq. (4), and substitute $\tau$ with Eq. (5), which leads to

$$\ln\left(\frac{R_{\mathrm{abs}}}{R_{\mathrm{ref}}}\right) = -c * \exp\left(-\frac{Z}{H}\right) * m \qquad (6)$$

15   Define

$$dbln\left(\frac{R_{\mathrm{abs}}}{R_{\mathrm{ref}}}\right) = \ln\left\{-\ln\left(\frac{R_{\mathrm{abs}}}{R_{\mathrm{ref}}}\right)\right\} \qquad (7)$$

We have

$$dbln\left(\frac{R_{\mathrm{abs}}}{R_{\mathrm{ref}}}\right) = \ln c - \frac{Z}{H} + \ln m \qquad (8)$$

Here *dbln* refers to the double logarithm, and the minus sign before the second logarithm function is added to avoid negative values. Eq. (8) decouples the effect of elevation and zenith angles in $dbln\left(\frac{R_{\mathrm{abs}}}{R_{\mathrm{ref}}}\right)$, which allows estimation of coefficients in Eq. (8) with simple multivariate linear regression using two independent terms $Z$ and $\ln m$:

$$dbln\left(\frac{R_{\mathrm{abs}}}{R_{\mathrm{ref}}}\right) \approx c_0 + c_1 Z + c_2 \ln m \qquad (9)$$

Here $c_0$, $c_1$, $c_2$ will be regression coefficients and can be used to predict the expected $dbln\left(\frac{R_{\text{abs}}}{R_{\text{ref}}}\right)$. Once $dbln\left(\frac{R_{\text{abs}}}{R_{\text{ref}}}\right)$ is solved, the $O_2$ band ratios can be derived with Eq. (10):

$$\frac{R_{\text{abs}}}{R_{\text{ref}}} = \exp\left(-\exp\left(dbln\left(\frac{R_{\text{abs}}}{R_{\text{ref}}}\right)\right)\right) \qquad (10)$$

The above derivation shows that the clear sky $O_2$ band ratios can be analytically predicted using surface elevation and zenith angles. Of course, many approximations have been used such as cancellation of Rayleigh extinction and surface BRF for the pair channels and constant absorption scale height. Due to large surface albedo, contributions of Rayleigh scattering are also

neglected. The contribution of Rayleigh scattering in the reflectance is about 0.01-0.02, and this may cause an uncertainty of 1% to 2% in the band ratio for bright surfaces. In cases of dark surfaces such as oceans, the surface albedo is so small (~0.05) that the Rayleigh scattering starts to dominate the observed reflectance, and the simple equations derived here will result in a large bias. However, with relatively large albedos (around 0.8), our sensitivity studies find the ratios

relatively stable, even though the single channel reflectances change in proportion to the surface albedo. The coefficients in Eq. (9) can be derived from either radiative transfer model simulations or real observational data from EPIC using multivariate least squares fitting. The advantage of the former is the exact knowledge of the model's atmosphere and clear or cloudy conditions. Conversely, its disadvantage is a limited number of atmospheric profiles and

sometimes simplistic or even unrealistic cloud input to the model. The advantage of using observational data is the abundant radiance measurements that could be used as a training dataset, while the disadvantage is the limited knowledge of atmospheric profiles and uncertainties in clear pixel identification. A common practice for developing a cloud mask algorithm is to use retrievals of simultaneous measurements from other better-equipped

instruments or ground observations as the truth. Exact same-time overpass is quite rare even with the vast data volume from the polar orbiting satellites such as Terra and Aqua, and cloud detection over snow and ice from instruments such as MODIS is itself subject to large uncertainty. This could lead to some false cloud/clear identification in the training dataset and bias the results. Based on the above reasoning, we first derive the $O_2$ band ratio thresholds with

both model simulations and observations, and then determine which set of coefficients is better suited for the EPIC cloud mask algorithm.

3. Radiative transfer simulations

3.1 Model setup

We used a radiative transfer simulator for EPIC (Gao et al., 2019) to generate the A-band and B-band reflectances over snow and ice surfaces. The EPIC simulator is built upon a radiative transfer model (Zhai et al., 2009, 2010) that solves multiple scattering of monochromatic light in the atmosphere and surface systems. Gas absorptions due to ozone, oxygen, water vapor, nitrogen dioxide, methane, and carbon dioxide are incorporated in all EPIC bands. The gas absorption cross sections are computed from the HITRAN line database (Rothman et al. 2013) using the Atmospheric Radiative Transfer Simulator (ARTS) (Buehler et al., 2011). Line broadening caused by pressure and line absorption parameters' dependences on temperature are considered. In the $O_2$ A- and B-bands, radiances from line-by-line radiative transfer simulations are convolved with EPIC filter transmission functions. The model atmosphere assumes a one-layer cloud with a molecular layer both above and beneath. The $O_2$ absorption within clouds is considered by assuming a fixed $O_2$ molecule vertical profile (US standard or other specified atmospheres).

For clear sky simulations, four atmospheric vertical profiles distributed with FASCODE (Chetwynd et al. 1994), originally from Intercomparison of Radiation Codes in Climate Models (ICRCCM) project (Barker et al. 2003), are used: 1976 US standard atmosphere, mid-latitude winter, subarctic summer and subarctic winter atmospheres. Surface albedo values used in the simulations are 0.6, 0.8 and 1.0 to represent snow or ice surface. The snow albedo varies from 0.5 to 0.9 depending on snow age, grain size, purity and sun angle, etc. (Warren, 1982) while ice albedo varies between 0.5 and 0.7. The daily mean snow albedo over Antarctica is generally over 0.8 (Pirazzini, 2004).

For cloudy sky cases, simulations for both water and ice clouds are conducted since both phases are found over the polar regions (e.g., Cesana et al. 2012, Zhao and Wang 2010). For water clouds, a gamma size distribution with effective radius of 10 µm and an effective variance

of 0.1 is assumed; for ice clouds, a fixed particle size (30µm) with a particle shape of severely roughened aggregate of hexagonal columns is assumed (Yang, et al., 2013). The cloud layer has varied optical thickness ranging from 0.2 to 30 and cloud top height from 1.0 km to 15 km above the ground. The cloud geometrical thickness varies from 0.5 km to 4 km.

The model simulates a variety of cases with 17 solar zenith angles ranging from 0° to 80°, 18 view zenith angles from 0° to 85°, and 37 azimuth angles from 0° to 180°, all with an increment of 5°. In addition to the varying sun-sensor geometry, the reflecting surface elevation is set from 0 to 7.5 km with a 2.5 km increment for the clear sky sensitivity tests while the cloudy sky simulations are performed at sea level and 2.5 km above sea level. See Table 1 for a complete list of the model parameters.

3.2 Clear sky simulations

We first examine whether the clear sky radiative transfer simulations are consistent with the simplified relationship between the $O_2$ band ratios and surface elevation and total airmass at typical surface albedo of 0.8 as discussed above (Eq. 9). A direct inspection of $O_2$ band ratios at a fixed view zenith angle and relative azimuth angles with surface elevation indicates a nearly linear relationship between the two (Fig. 1a, 1b). The relationship depends on the solar zenith angle. At a higher solar zenith angle, not only are the ratios lower at all surface elevations but also the rate of change with height ($\frac{\partial r}{\partial Z}$) is larger. However, the same relationship can be expressed as a quasi-linear relationship between Z and the double logarithm of $O_2$ band ratios at fixed zenith angles as indicated by Eq. (9) (Fig. 1c, 1d).

The variation of $O_2$ band ratios with solar zenith angles has been discussed in previous works (Fischer, J. and Grassl, 1991; Wang et al. 2008; Yang et al., 2013; and Gao et al., 2019). Here we show a more quantitative dependence of $O_2$ band ratios as a function of the total relative airmass ($m$) defined in Eq. (3) at fixed surface elevation (sea level in this case, Fig. 1e, 1f). The inverse relationship of $O_2$ band ratios with $m$ is evident. Although EPIC is positioned close to the backscattering direction, there is a small difference in $\theta_s$ and $\theta_v$, generally smaller than 6°. The red dots show the simulations when the difference between $\theta_s$ and $\theta_v$ is smaller than 6° to mimic

the EPIC sun-view geometry. The relationship derived from samples with restricted view zenith angles is not much different from that of all samples. Figures 1g-h further project this relationship as logarithm of $m$ versus double logarithm of $O_2$ band ratios as shown in Eq. (9). We notice that the linear relationship holds very well except for very large relative airmass ($\ln(m) >$ 2.5, which corresponds to zenith angles $> 80°$).

To account for both elevation and zenith angle effect, a multivariate least square regression is applied in which $Z$ and $\ln(m)$ are taken as two independent terms and $dbln$ $(\frac{R_{abs}}{R_{ref}})$ is the dependent variable for the simulations, as suggested in Eq. (9), with the sample restricted to a zenith angle difference of below 6°. The results indicate high confidence of the fitting, with multi-correlation coefficients reaching 0.998 for both A-band and B-band simulations (Fig. 1i, 1j). The coefficients $c_0$, $c_1$, and $c_2$ are listed in Table 2. The set of regression coefficients derived from simulations at surface albedo 0.8 also predict very well the A-band ratios from simulations using different surface albedos (0.6 and 1.0) (Fig. 2a), with obvious divergence occurring only at large zenith angles (>80°) where no retrieval is performed for EPIC (Fig. 2b).

Table 2 also lists the set of coefficients derived from observations utilizing information from collocated GEO/LEO pixels. Details will be discussed in Section 4.

3.3 Cloudy sky simulations

The coefficients in Table 2 can be applied to Eq. (9) to compute an expected clear sky band ratios. In order to test the feasibility of using the derived clear sky band ratios as the thresholds for clear and cloudy pixel separation, we first evaluate the sensitivity of $O_2$ band ratios to cloud properties. This is done by adding clouds with different optical thickness, cloud top height and geometric thickness in the radiative transfer simulations, and then comparing the $O_2$ band ratios of cloudy sky with those of clear sky under the same sun-view geometry. The results from solar and view zenith angles of 30° and 60° and relative azimuth angle of 160° are shown in Figure 3, with the corresponding clear sky values shown as the filled and open triangles, respectively. We notice that the $O_2$ band ratios generally increase with the optical thickness and are higher for cloudy skies than for clear skies but with certain exceptions. At low zenith angles (< 30°), we

find very low sensitivity of $O_2$ band ratios with cloud optical thickness when cloud top height is 1 km (Fig. 3a, 3b). Likewise, the sensitivity to cloud top height is very low at low optical thickness (tau = 1.7) for the A-band (Fig. 3c). For the B-band, the $O_2$ ratios decrease with cloud top height up to 5 km before increasing again at tau = 1.7 (Fig. 3d). Note that these figures show

that adding a layer of optically thin cloud (COT < 3) actually decreases the ratio at 30° zenith angle. The reason is that under this circumstance the reflectance of the reference channel increases more than the absorption channel, which indicates an increase in the photon path. The causes of photon path increase include multiple scattering inside the cloud and surface-cloud interaction. The strong surface-cloud interaction over the bright surface of snow and ice partly

contributes to the low sensitivity of $O_2$ band ratios for the low and thin clouds compared with relatively darker surfaces (Further illustrated in Fig. 4). The sensitivity of $O_2$ band ratios to cloud optical thickness and height increases with solar and view zenith angles, as can be seen from the SZA = VZA = 60° curves.

As the cloud mask only works when cloudy sky $O_2$ band ratios are greater than the clear sky ratios, the difference between the two at low zenith angles (vza = sza = 30°) is shown as a function of two major factors: COT and CTOP for the A-band and B-band at surface albedo 0.8, cloud geometric thickness of 1km and sea level conditions (Fig. 4a, 4b), along with their sensitivities with altered geometric thickness (Fig. 4c, 4d), surface albedo (Fig. 4e, 4f), and

surface elevation (Fig. 4g, 4h). If a difference larger than 0.01 is required to confidently detect cloud, we notice that the cases at the lower left side of each figure, which correspond to low COT and CTOP, will present difficulty in cloud detection. Smaller cloud geometric height (Fig. 4c, 4d) and surface albedo (Fig. 4e, 4f) tend to increase the sensitivity while higher surface elevation (Fig. 4g, 4h) tends to decrease the sensitivity as compared to the cases in Fig. 4a and 4b

for the A-band and B-band, respectively. These results show that $O_2$ band ratios can be used to detect clouds that are thick and/or high with much confidence over snow and ice surfaces. Difficulties still exist in detecting thin clouds or low clouds at low zenith angles (<30°). Note that the A-band has better sensitivity than the B-band, as expected. It should be pointed out that for most of the cases, the solar zenith angles are larger than 30° since snow and ice are present

mainly in regions of high latitudes.

4. EPIC cloud mask over snow and ice surfaces

The regression results from Eq. (9) can be used as the thresholds for cloud detection. As discussed in Section 2, we can derive the thresholds using either radiative transfer simulations or satellite observations. The previous section discussed the path of using modeling results, here we attempt to derive the thresholds based on the real EPIC data.

For this purpose, the Langley GEO/LEO composite cloud product (Khlopenkov et al., 2017) and EPIC L1B data from January and July of 2017 are used as the training dataset, and data from January and July 2016 are used for validation. The cloud retrievals in the composite data follows Minnis et al. (2011). Because of EPIC's large pixel size, one EPIC pixel corresponds to many GEO/LEO pixels each with its own cloud mask and optical properties retrievals, hence a composite pixel reports a cloud fraction based on cloud masks of the GEO/LEO pixels within it. It should be noted that cloud detections over snow and ice surfaces from instruments on GEO/LEO satellites are difficult as well. For example, the AVHRR-based cloud fraction was found to be basically unbiased over most of the globe except over the polar regions where a considerable underestimation of cloudiness could be seen during the polar winter when compared with cloud information from the Cloud-Aerosol Lidar with Orthogonal Polarization (CALIOP) onboard the CALIPSO satellite. The overall probability of detecting clouds in the polar winter could be as low as 50 % over the highest and coldest parts of Greenland and Antarctica, with a large fraction of optically thick clouds remaining undetected (Karlsson et al. 2018). Wang et al. (2016) shows MODIS from Terra and Aqua misidentifies cloud as clear as high as 20% over snow covered or sea ice regions in Antarctica. They show that misidentification of clear as cloud also occurs quite frequently in Eastern Antarctica during boreal spring and fall. Over snow covered high mountains over the Tibetan Plateau, a recent study by Shang et al. (2018) found the cloud detection rate to be 73.55% and 80.15% for the Advanced Himawari Imager (AHI) and MODIS, respectively. All these studies use the CALIOP cloud detection as ground truth and highlight the large uncertainties in cloud detection from passive radiometers over snow and ice surfaces and over high mountain areas.

Keeping these in mind, we use the GEO/LEO composite cloud product as the training and validation dataset because of its pole-to-pole coverage and availability. The cloud fraction and surface scene types from the composite dataset are used to select the clear pixels (100% clear) over snow and ice surfaces (when 90% of the scene type is permanent snow or ice, seasonal

snow, or ice over water). Surface type is reported in the Langley GEO/LEO dataset, which is based on the IGBP surface type dataset and the Near-real-time Ice and Snow Extent (NISE) data set from the National Snow & Ice Data Center (NSIDC) (Brodzik and Stewart, 2016). To reduce the uncertainties, we further restrict the difference between the GEO/LEO and the EPIC to be within 5 minutes. We also restrict the analysis on pixels with view zenith angle less than 80°.

The surface elevation data is from the National Geophysical Data Center (NGDC) TerrainBase Global Digital Terrain Model (DTM), version 1.0 (Row and Hastings, 1994).

The same type of multivariate least square regression is performed for the clear sky pixels using the elevation and logarithm of total relative airmass as independent variables, and the double logarithm of the $O_2$ band ratios as the dependent variables as suggested by Eq. (9). The

derived regression coefficients (Table 2) are quite close to those derived from the model simulations with slightly larger scatter (Fig. 5a, 5b). One major source of uncertainty may come from the GEO/LEO cloud identification. As mentioned above, cloud detection over snow and ice surfaces is very challenging even for GEO/LEO satellites with more spectral channels. Cloud contaminated pixels might have lower or higher $O_2$ band ratios than the clear sky values

depending on the optical thickness of the cloud and the sun/viewing geometry (Fig. 3).  Other sources of uncertainties, such as geolocation, surface elevation and atmospheric profile can also contribute to the larger scatter in the observational data.

Obviously, the clear sky thresholds predicted from observational data must be adjusted to provide a better overall performance since the regression model is designed to predict the mean rather than the upper bound of clear sky band ratios. The same regression coefficients applied to

cloudy sky samples indicate many overlapping of $O_2$ band ratios from clear sky and cloudy sky pixels (Fig. 5c, 5d). A threshold value too high will guarantee the clear sky identification but underestimate cloudy pixels, and too low will lead to overestimation of cloudy pixels. To achieve the best overall clear sky and cloudy sky performance, i.e., a balanced correct detection rate and

false detection rate as discussed in Section 5, we set the threshold value by increasing the ratios

derived from Eq. (10) by 0.025 so that the cloud mask threshold is close to the upper quantile of the clear sky values (red dashed line in Fig. 5c and 5d).

Results show that using the set of coefficients derived from the model simulations captures most of the clear sky samples without being adjusted (Figures not shown). We found that even though the thresholds derived from the observational data perform slightly better when applied back to the same training dataset, they underperform the model derived algorithm when applied to a different data period (January and July of 2016). One likely reason is that the cloud identification in the observational training dataset has its own non-negligible uncertainties. These uncertainties will not affect the performance in the training dataset but affect the algorithm performance in a different data period. For this purpose, we adopt the algorithm derived from the model simulations for the rest of this paper.

Following the current EPIC cloud mask algorithm, we also set an upper and a lower threshold that is 0.02 above or below the model predicted threshold ($RT_0$). A cloud mask (CM) confidence level is determined for each pair of the $O_2$ band ratios based on whether the ratios fall between these intervals/thresholds:

$$
CM = \begin{cases}
4 & \text{Ratio} > RT_0 + 0.02; & \text{CldHC} \\
3 & RT_0 < \text{Ratio} < RT_0 + 0.02; & \text{CldLC} \\
2 & RT_0 - 0.02 < \text{Ratio} < RT_0; & \text{ClrLC} \\
1 & \text{Ratio} < RT_0 - 0.02; & \text{ClrHC}
\end{cases}
$$

Here, CldHC, CldLC, ClrHC, and ClrLC refer to Cloud with High Confidence, Cloud with low Confidence, Clear with High Confidence; Clear with low Confidence, respectively. The final confidence level is determined by combing the two results from the A- and B-band tests according to Table 3. Note that we only define high confidence cloud (CldHC) or high confidence clear (ClrHC) when both tests show cloud or clear with high confidence.

An illustration of EPIC $O_2$ band ratios and the derived cloud mask over the Antarctic on Dec.23, 2017 is shown in Figure 6, along with cloud fraction derived from GEO/LEO composite. In this figure, the A-band and B-band ratios show not only the presence of clouds but also the effect of elevation, as the low values over Ross Ice Shelf are clearly influenced by the low

elevation in that area. The new cloud mask detects the majority of the cloud area, but some portion of clouds over this region is missing. This could be because the clouds in this scene over the Ross Ice Shelf are low.

5        5.   Algorithm validation

Using the thresholds from radiative transfer simulations, we reprocessed the EPIC cloud mask over snow and ice surfaces for all the collocated pixels in three months: January 2016, January 2017, and July 2017.

We divide the GEO/LEO cloud fraction into 4 categories to match with the CM in EPIC:

$$\text{GEO/LEO CM} = \begin{cases} 4: & \text{cloud fraction} \geq 95\% \\ 3: & 50\% \leq \text{cloud fraction} < 95\% \\ 2: & 5\% \leq \text{cloud fraction} < 50\% \\ 1: & \text{cloud fraction} < 5\% \end{cases}$$

15        Figure 7 shows the 4 x 4 fusion matrixes of the EPIC cloud mask with the GEO/LEO cloud fraction for the three months. The diagonal squares represent agreement between the GEO/LEO and EPIC cloud masks, while the off-diagonal squares represent disagreement between the two products. For January 2016 and 2017, we notice that the original algorithm has a high percentage of pixels in the left-bottom corner (clear – clear) category,  but there is a large percentage of
20   GEO/LEO cloudy pixels in the >95% category miss-identified by EPIC as clear (cloud mask = 1). There are also a considerable amount of pixels in the low GEO/LEO cloud fraction category (<5%) being classified as cloudy (CM = 3, 4). Improvement is evident for the new algorithm, where percentages of pixels in clear - clear (< 5% and CM = 1) and cloudy - cloudy  (>95% and CM = 4) are significantly increased. The changes in July 2017 are less obvious, as the original
25   algorithm already captures large percentage of pixels in clear -clear and cloudy-cloudy categories.

To quantitatively measure the performance of the cloud masking algorithms, we further define a binary partition of Negative (CM = 1, 2, or cloud fraction <5% and 5-50%) and Positive (CM = 3, 4, or cloud fraction 50-95% and >95%) cloud identification for both EPIC and

GEO/LEO, which results in 4 total combinations. Successful retrievals consist of TP (True Positive) and TN (True Negative) cases, in which both algorithms identify the pixel as cloudy and clear, respectively, and unsuccessful retrievals consist of FN (False Negative) and FP (False Positive) – where EPIC identifies a pixel as clear and cloudy respectively, opposite to GEO/LEO cloud mask. Assuming GEO/LEO is the "truth," a number of parameters as a measure of EPIC's CM accuracy are computed:

$$Accuracy = \frac{TP + TN}{TP + TN + FN + FP} \qquad (11)$$

$$POCD = \frac{TP}{TP + FN} \qquad (12)$$

$$POFD = \frac{FP}{TN + FP} \qquad (13)$$

Here POCD and POFD are the probability of correct detection and probability of false detection, respectively. For January 2016 and 2017, compared to the current product, the accuracies have been improved considerably from a low 57-60% to around 83%. The POCD is nearly doubled (from 36% to 64-67%) and a significant reduction of POFD (a drop from around 50% to 10%). The original algorithm performs relatively well in July 2017 with a probability of correct detection (POCD) at 77.5% and a low probability of false detection (POFD) of 16.5%; hence the improvement for this month is relatively small.

Figure 8 shows the cloud fraction on a $1_\circ$ x $1_\circ$ grid for January 2017 over snow and ice covered Antarctica.  Note that here we lift the 5 min time difference limitation and use all available pixels with view zenith angles less than 75° from the GEO/LEO composites (Khlopenkov et al., 2017) in order to have a full coverage of the region. The cloud fraction map from GEO/LEO shows a belt of high cloud fraction originated from mid-latitude storm track reaching the edge of the continent. Onto the icy plateau of East Antarctica, cloud fraction quickly decreases.  High cloud fraction is found over West Antarctica.  The cloud fraction from the original algorithm shows quite an opposite cloud distribution pattern between West and East Antarctica. This is likely due to fixed threshold that is too low for the high elevation in East Antarctica and too high for the low elevation in West Antarctica. By taking the elevation into

account, the new algorithm identifies the regional cloud distribution much better. In addition, the new algorithm also has a better cloud fraction match around the edge of the Antarctic continent.

To examine the performance of the new algorithm on the global scale, we plotted gridded cloud fraction over snow and ice surfaces for the entire globe in January 2016 (Fig. 9). The number of snow/ice pixels used for the map are also shown, because sample numbers affect the quality of monthly mean. We notice that the number of snow/ice pixels per grid is much higher in January over Antarctica. There are also considerable amounts of snow/ice pixels in northern hemisphere high-latitude regions and the southern tip of the Andes. There is no retrieval north of 50° N due to no daylight or view zenith angle too large in January (DSCOVR only has observations for the daytime Earth). Comparisons show that the new algorithm improves cloud distributions noticeably.

Figure 10 shows a similar map but for July 2017. During the boreal summer, the cloud mask algorithm has retrievals over the entire northern hemisphere but not for part of Antarctica south of 65 °S due to the polar night. The GEO/LEO cloud fraction map indicates cloud fraction > 80% over snow and ice surfaces over most of the regions in July except over Greenland. The original algorithm has similar cloud fraction in most areas over snow and ice surfaces, except over southeast Greenland where it has significantly more cloud than the other part of Greenland. This is likely due to the original algorithm's failure to take into consideration the high elevation there. On the other hand, the underestimation of cloud fraction at the southern tip of the Andes could be due to its failure to take into account the large solar and view zenith angles in summer. The new algorithm detects significantly lower amount of cloud fraction in Greenland and improves the cloud detection in the aforementioned high mountain areas.

Even though the new cloud mask has improved the accuracy and general distribution compared with the GEO/LEO retrievals, regional differences between the two can still be quite large. This is partly due to the large uncertainty of cloud detection from GEO/LEO over snow/ice itself, and partly due to the intrinsic difficulty of using $O_2$ band ratios in detecting the low cloud and thin cloud as discussed before. In addition, the time difference between EPIC and GEO/LEO observations can also impact the comparison between the two. Stratifying the performance based

on difference in the observation time, we find a larger difference in the observing time leads to slightly lower POPD, higher POFD and an overall decreasing accuracy (Fig. 11).

## 6. Summary and discussion

Due to limited spectral channels, especially the lack of infrared and near infrared channels in the DSCOVR EPIC instrument, cloud detection for EPIC over snow and ice poses a great challenge. The existing EPIC cloud mask algorithm employs two oxygen pair ratios in A-band (764 nm, 780 nm) and B-band (688 nm, 680 nm) for cloud detection over the snow and ice surfaces. This method is based on the fact that photons reflected by clouds above the surface will travel, on average, a shorter distance through the atmosphere and so experience less absorption by $O_2$; hence a threshold can be set to separate cloudy pixels from clear pixels. However, clear sky $O_2$ band ratios depend on a number of factors such as surface elevation and sun/viewing geometry that impact the total absorption airmass; these factors need to be accounted for.

In this study, we use both the radiative transfer theory and model simulations to quantify the relationship between the $O_2$ band ratios with surface elevation and zenith angles. Thresholds are derived as a function of surface elevation and sun-view geometry based on both model simulation results and observations. The model derived algorithm is chosen because it performs better when applied to the observations that were not used in the training dataset. The new algorithm increases the accuracy of the EPIC cloud mask over snow and ice surfaces in winter by more than 20%. This is achieved through a significant reduction of false detection rate from 50% to 10% and nearly doubling of the correct detection rate. The improvement in July is mild, with the main improvement observed over Greenland. Of course, these performance metrics are based on comparison with GEO/LEO cloud mask which has quite large uncertainty over snow and ice surfaces itself. In addition to significant improvement in cloud detection over Antarctic, the new algorithm also improves cloud detection over Greenland and some mid-latitude high mountain areas.

Limitations of this method include difficulties in identifying thin cloud with optical thickness less than 3 or low cloud below 3 km due to the lack of sensitivity in $O_2$ band ratios

under these circumstances. Compared with the infrared-based techniques, one advantage of this oxygen band technique is that it is relatively insensitive to the surface and atmosphere temperature. Therefore, the method presented in this work provides a solution to polar cloud detection when infrared channels are not available, or struggle to distinguish between cloudy and

5    clear scenes.  We anticipate that cloud detection using oxygen band technique to be of great value in the future missions.

**Acknowledgments**

We thank the two anonymous reviewers for their thorough and very helpful reviews. This

10    research was supported by the NASA DSCOVR Earth Science Algorithms program managed by Richard Eckman. The DSCOVR level-1 and level-2 data used in this paper are publicly available from NASA Langley Atmospheric Sciences Data Center (ASDC). The ice cloud optical properties database is obtained from Professor Ping Yang at the Texas A&M University.

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

Table 1: Parameter setup in radiative transfer model simulations

| | | |
|---|---|---|
| Clear Sky Simulations | Atmospheric Profiles | Standard US 1976; Mid-Latitude Winter; Sub-Arctic Summer; Sub-Arctic Winter; |
| | Solar Zenith Angles | 0-80°, every 5° |
| | View Zenith Angles | 0-75°, every 5 |
| | Relative Azimuth Angles | 0-180°, every 5° |
| | Surface Elevation | 0.0, 2.5, 5.0, 7.5 km |
| | Surface Albedo | 0.8, 0.6, 1.0 |
| Cloudy Sky Simulations | Atmospheric Profiles | Standard US 1976 |
| | Solar Zenith Angles | 0-80°, every 5°, (30°, 60° for surface elevation = 2.5 km and surface albedo = 0.6) |
| | View Zenith Angles | 0-75°, every 5 |
| | Relative Azimuth Angles | 0-180°, every 5° |
| | Surface Elevation | 0, 2.5 km |
| | Cloud Top Height | 1.00, 3.00, 5.00, 7.50, 10.00, 12.50, 15.00 km |
| | Cloud Geometric Thickness | 0.50, 1.00, 2.00, 4.00 km |
| | Cloud Optical Thickness | 0.22, 0.82, 1.72, 3.06, 5.05, 8.03, 12.46, 19.09, 28.96 |
| | Surface Albedo | 0.8, 0.6 |

Table 2. Regression coefficients for equation (9) and multiple correlation coefficients (Rmulti) derived from model simulated data and observations, respectively.

| | A-band | | | | B-band | | | |
|---|---|---|---|---|---|---|---|---|
| | $c_0$ | $c_1$ | $c_2$ | Rmulti | $c_0$ | $c_1$ | $c_2$ | Rmulti |
| Simulations | -0.3100 | -0.1341 | 0.5202 | 0.998 | -1.0201 | -0.1361 | 0.4888 | 0.999 |
| Observations | -0.1764 | -0.1152 | 0.4542 | 0.958 | -0.8672 | -0.1185 | 0.3995 | 0.934 |

Table 3. The logic table for combining the cloud mask results from the A- and B-band tests. Acronyms CldHC: Cloud with High Confidence; CldLC: Cloud with low Confidence; ClrHC: Clear with High Confidence; ClrLC: Clear with low Confidence.

| | | A-band Test | | | |
|---|---|---|---|---|---|
| | | CldHC | CldLC | ClrLC | ClrHC |
| B-band test | CldHC | **CldHC** | CldLC | CldLC | CldLC |
| | CldLC | CldLC | CldLC | CldLC | ClrLC |
| | ClrLC | CldLC | CldLC | ClrLC | ClrLC |
| | ClrHC | CldLC | ClrLC | ClrLC | **ClrHC** |

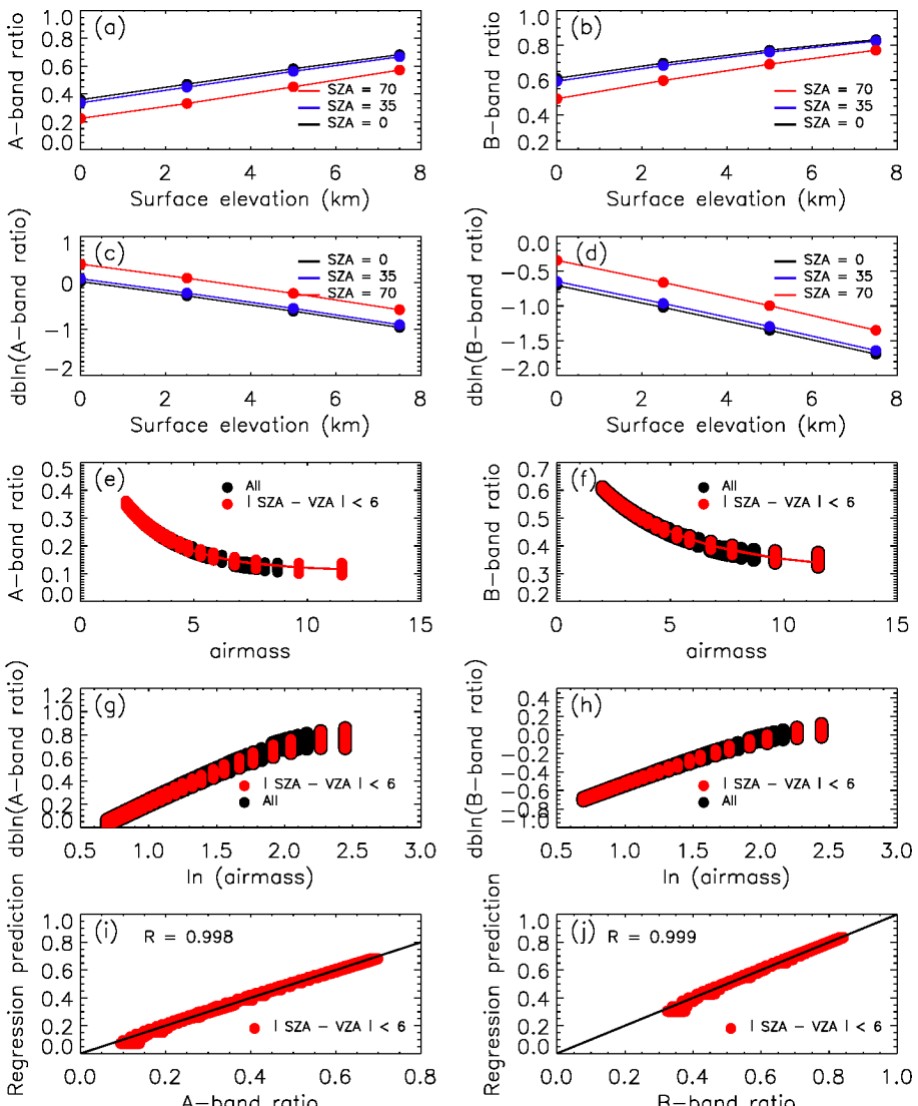

Figure 1. Relationships between model simulations of clear sky A-band (left column) and B-band (right column) ratios with surface elevation and relative airmass. a, b) $O_2$ band ratios as a function of surface elevation; c, d) double logarithm of $O_2$ band ratios versus surface elevation; e, f ) $O_2$ band ratios as a function of total relative airmass; g, h) double logarithm of $O_2$ band ratios versus logarithm of total relative airmass; i, j) scatter plot of fitted thresholds and $O_2$ band ratios. The red points in Panels e-j show the simulations when the difference between $\theta_s$ and $\theta_v$ is smaller than 6° to mimic the EPIC sun-view geometry. The fitted thresholds are computed with a multivariable linear regression in which double logarithms of $O_2$ band ratios are expressed as a function of surface elevation and logarithmic of total relative airmass. The simulations use 4

atmospheric profiles: mid-latitude winter, subarctic summer, subarctic winter, standard US atmosphere. Surface albedo is set at 0.8 to represent snow and ice surface.

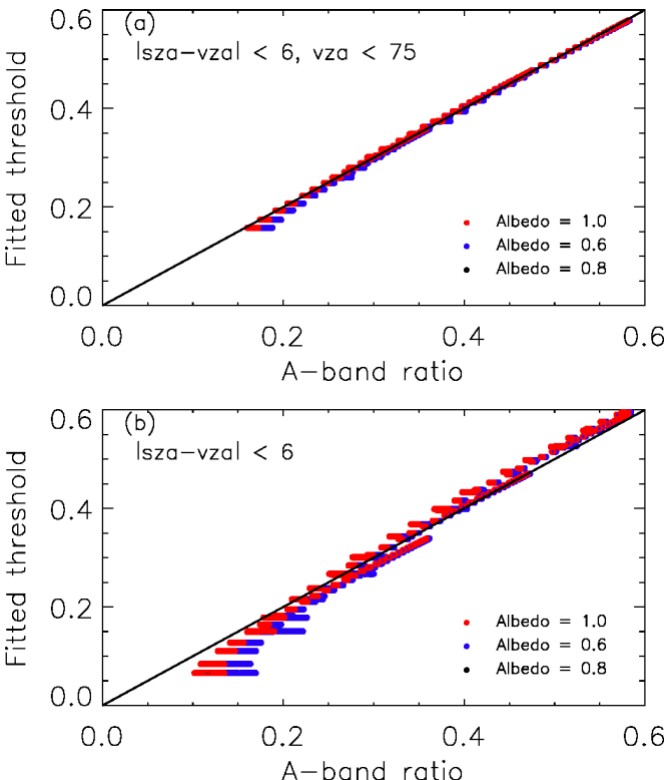

5    Figure 2. Scatter plot of model simulated A-band ratios (y-axis) at surface albedo = 0.6 (blue), 0.8 (black) and 1.0 (red) versus computed with regression derived with the set of simulations at surface albedo = 0.8 (x-axis) for (a) view zenith angles < 75°, and (b) all view zenith angles. Absolute solar zenith angle and view zenith angles differences are smaller than 6° for both plots. The results are from simulations using standard US atmosphere.

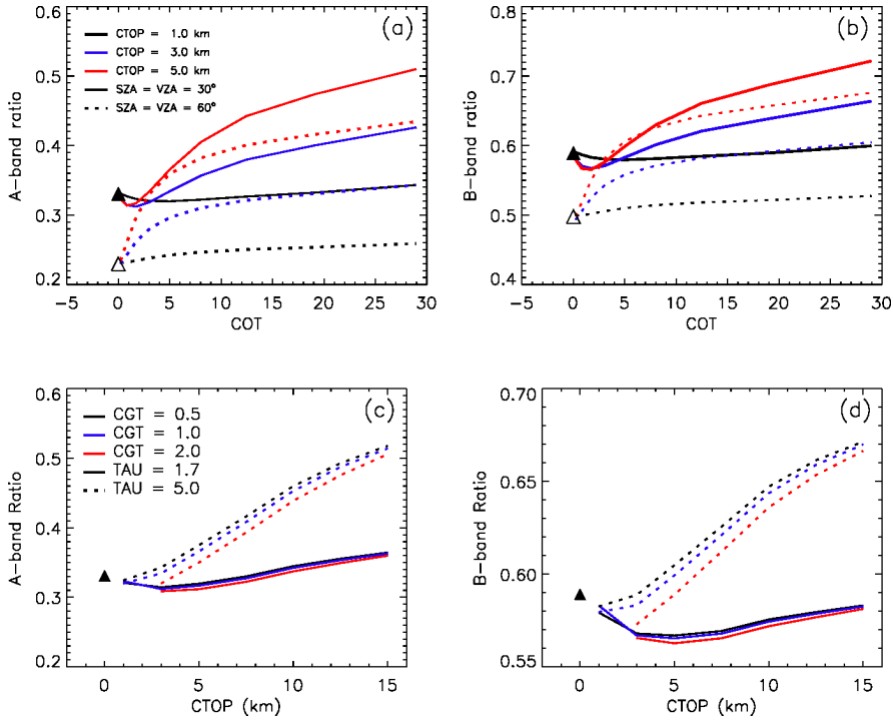

Figure 3. Model simulated Oxygen band ratios as a function of cloud optical thickness (COT) with cloud top height at 2.5 km (black), 5.0 km (blue) and 7.5 km (red) and solar zenith angles at 30° (solid line) and 60° (dotted lines), respectively for (a) A-band and (b) B-band. View zenith angle is the same as the solar zenith angle and relative azimuth angle is 160° for all the simulations. The clear sky simulations are marked with filled and unfilled triangles for solar and view zenith angles at 30° and 60°, respectively. Both clear sky and cloudy sky simulations use standard US atmosphere and zero ground elevation. Relative Azimuth Angle is 160. Surface albedo is set at 0.8 to represent snow and ice surface.

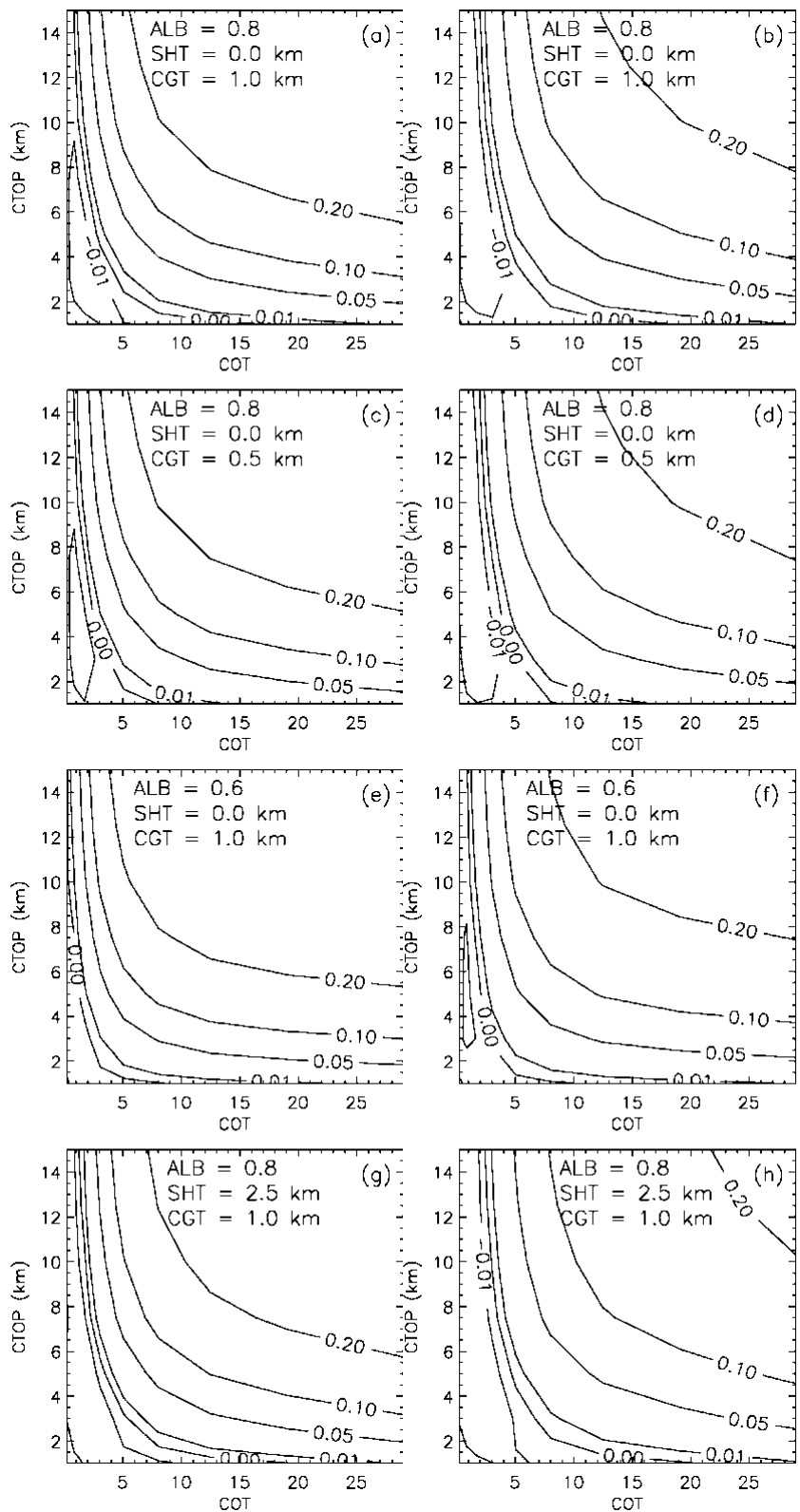

Figure 4. The difference of O$_2$ band ratios (cloudy sky - clear sky) as a function of COT and

CTOP at SZA = VZA = 30°, RAZM = 160° at (a, b) surface albedo (ALB) = 0.8, surface height

(SHT) = 0 km (sea level), and cloud geometric thickness (CGT) = 1 km; the rest are the same as (a, b), but with the change of one parameter for (c, d) CGT = 0.5 km ; (e, f) ALB = 0.6 ; and (g, h) SHT = 2.5 km.  The right panel is for A-band and the left panel is for B-band.

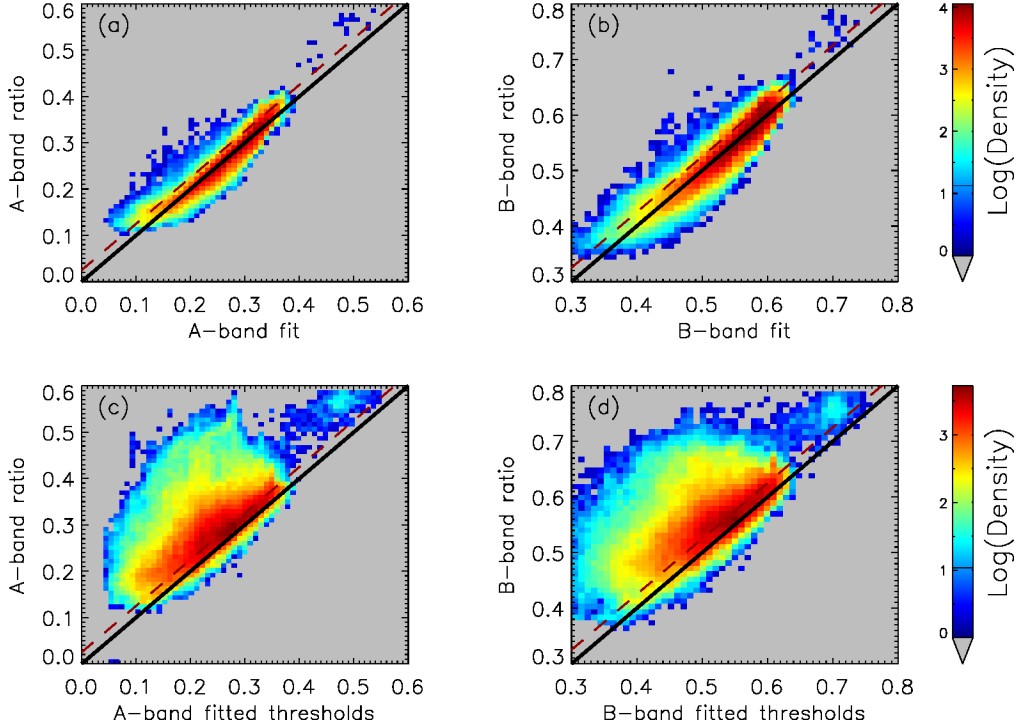

Figure 5. Scatter plot of regression fit versus A-band (left) and B-band (right) ratios for clear sky (a, b) and cloudy sky (c, d) pixels from EPIC measurements over global snow and ice surfaces in January and July 2017. The regression is derived with clear sky oxygen band ratio as a function of surface elevation and airmass. The pixels on the left (right) side of black lines could be identified as cloudy (clear) as the observed ratios is larger (smaller) than the predicted threshold. The dashed lines (increase the predicted ratios by 0.025) provide better division of clear and cloudy pixels.

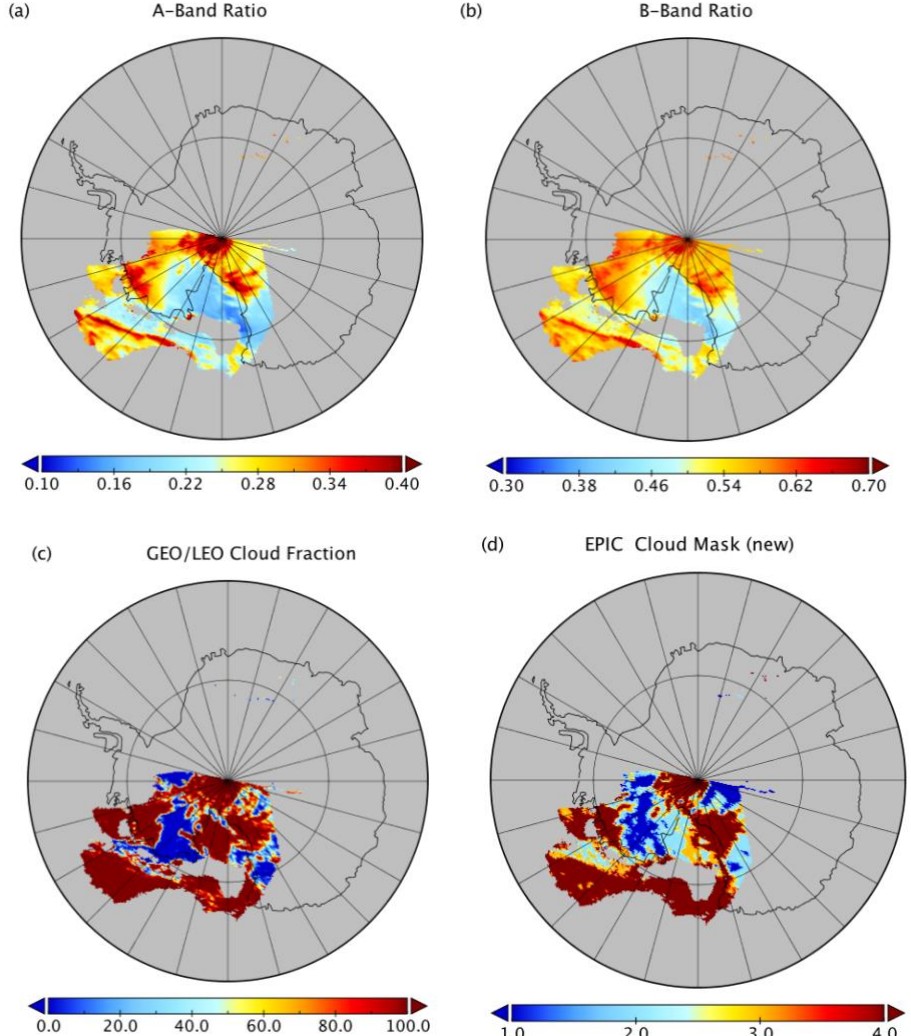

5    Figure 6. Section of an EPIC granule on Dec 23, 2017, 1707 UTC time with matching GEO/LEO
overpass within 5 minutes of the EPIC scan over western Antarctic. (a) A-band ratio, (b) B-band
ratio, (c) cloud fraction from GEO/LEO composite, (d) Cloud mask from the new algorithm.

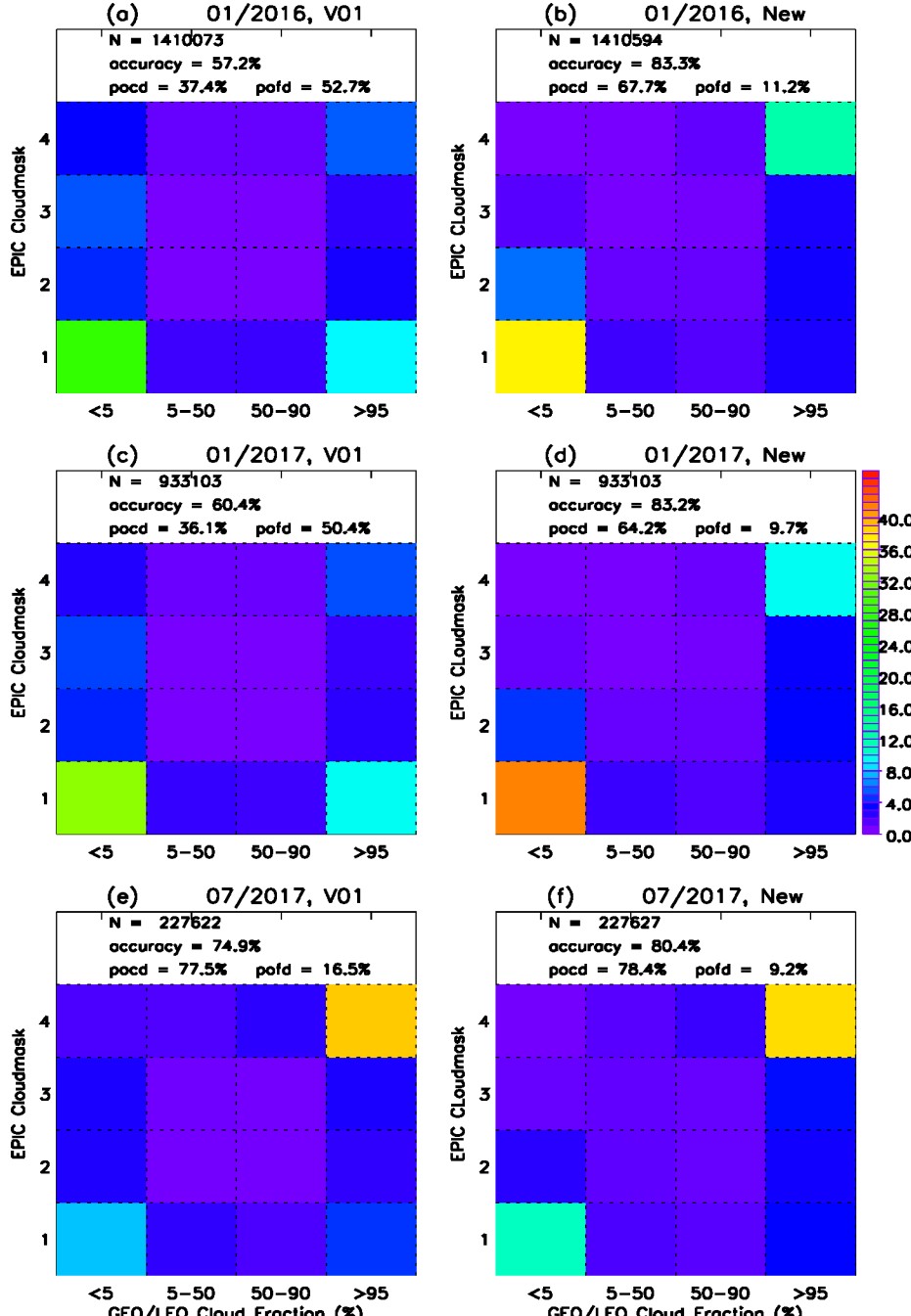

Figure 7. Percentage of pixels in each pixel-by-pixel matchup category between cloud mask from EPIC and GEO/LEO composite cloud fraction over snow and ice surfaces for January 2016 (a, b), January 2017 (c, d), and July 2017 (e, f). Left is from the current EPIC cloud mask algorithm and the right is from the new algorithm. The diagonal squares represent agreement between GEO/LEO and EPIC cloud mask, while the off-diagonal squares represent disagreement between

the two products. The number of samples, accuracy, probability of correct detection (POCD), and probability of false detection (POFD) are shown in the white area on top of each figure.

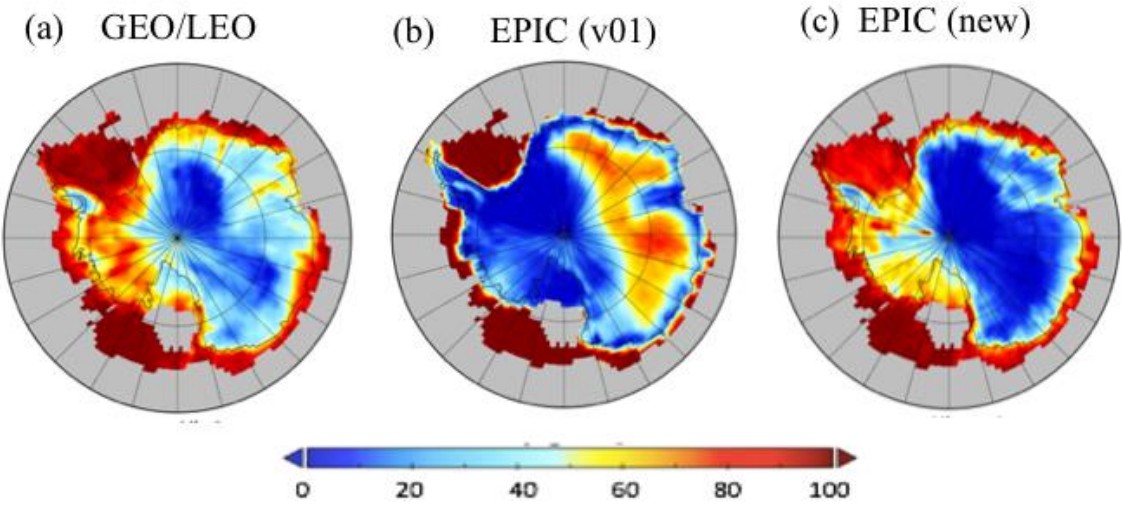

Figure 8. Cloud Fractions derived from (a) composite GEO/LEO retrievals, (b) original EPIC cloud mask, (c) new EPIC cloud mask over Antarctic in January 2017.

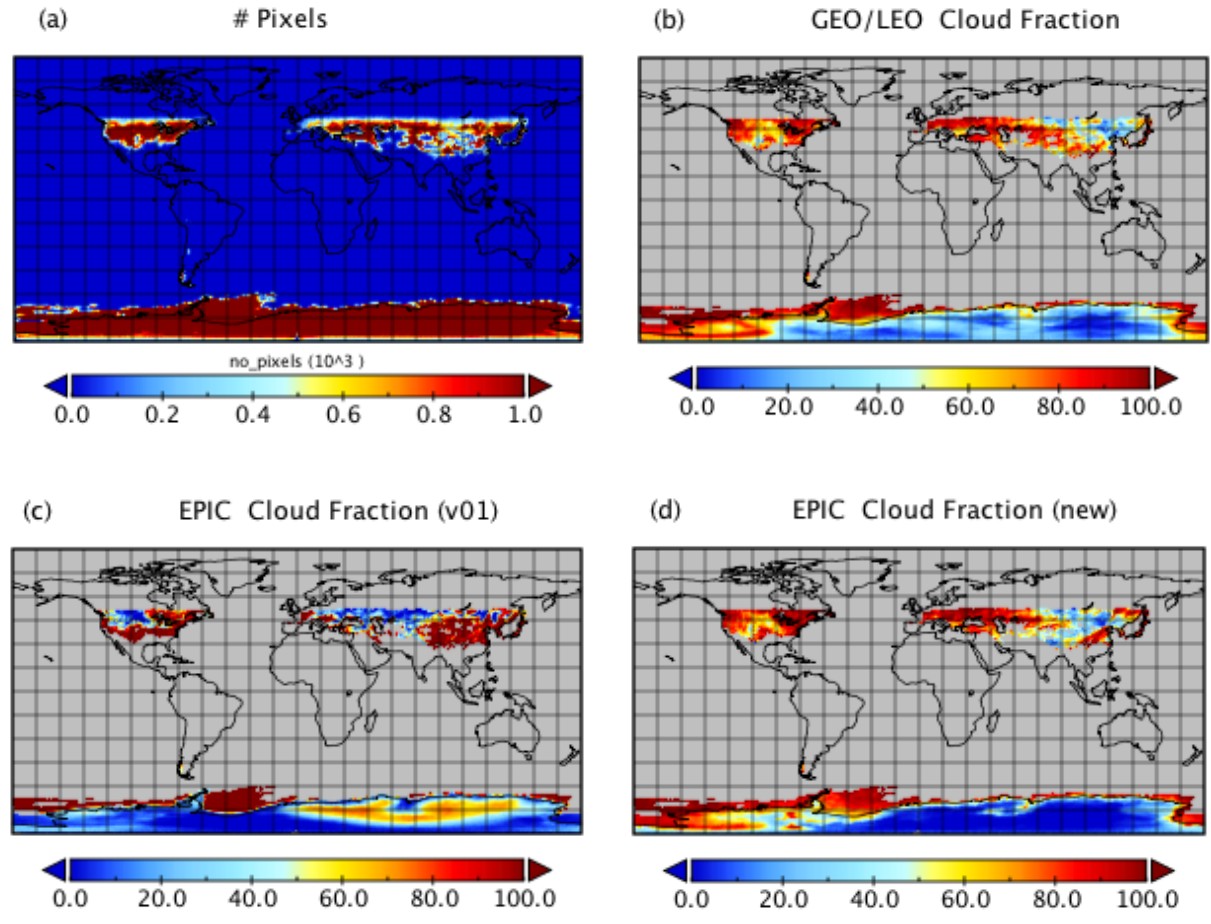

Figure 9. (a) Number of ice/snow pixels and monthly mean cloud fractions derived from (b) GEO/LEO composites, (c) original EPIC cloud mask algorithm, and d) new algorithm in 1° x 1° grids for January 2016.

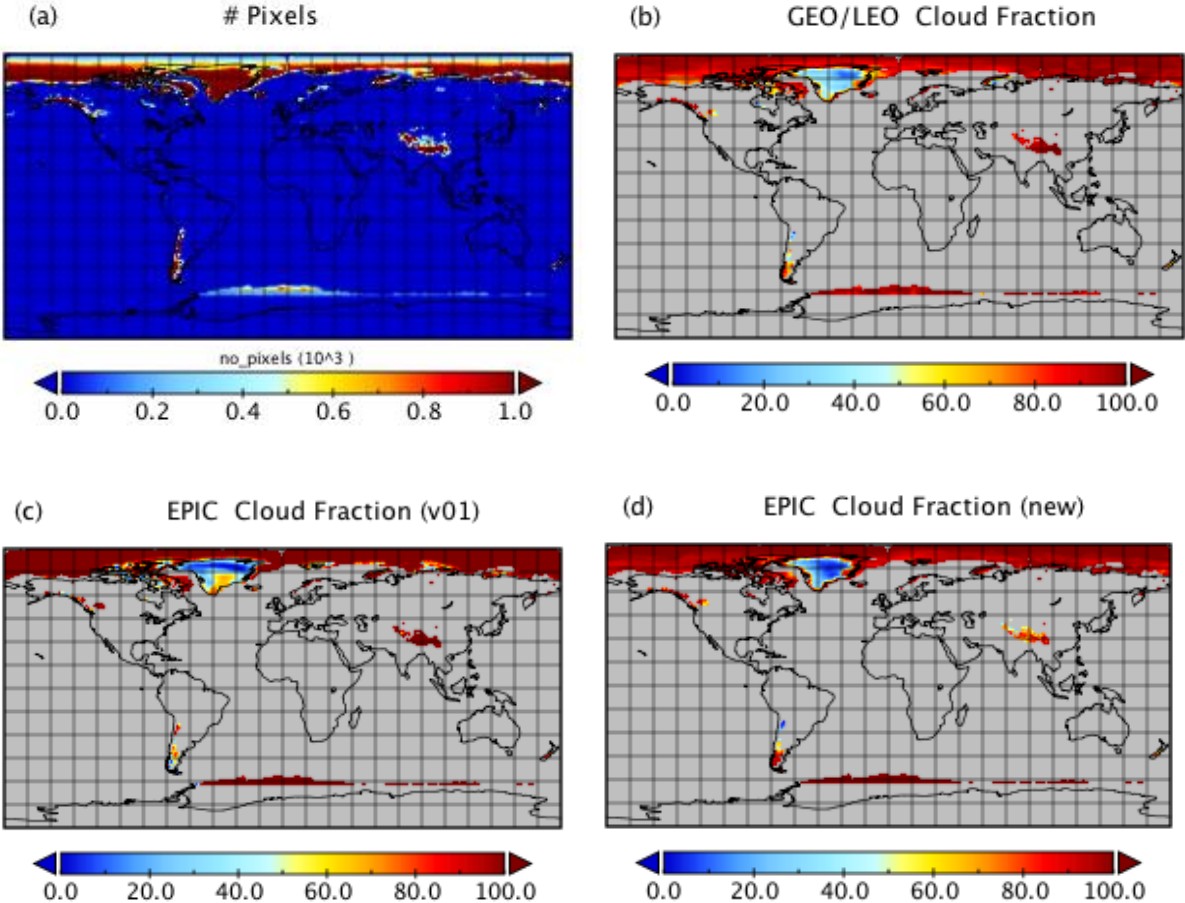

5    Figure 10. (a) Number of ice/snow pixels and monthly mean cloud fractions derived from (b) GEO/LEO composites, (c) original EPIC cloud mask algorithm, and d) new algorithm in 1° x 1° grids for July 2017.

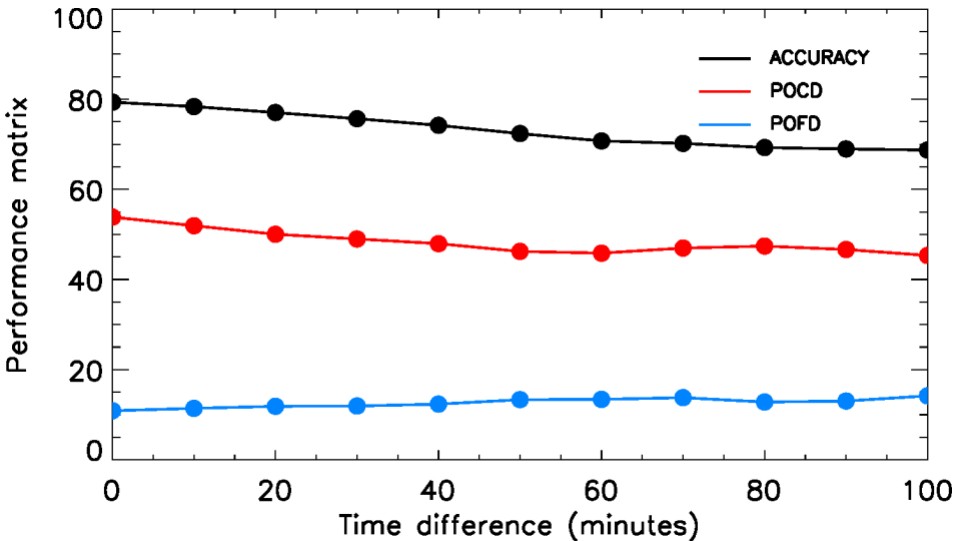

5      Figure 11. Performance metrics for January 2017 as a function of time difference between EPIC
       and GEO/LEO instrument measurements. POCD: probability of correct detection; POFD:
       probability of false detection.