# Peer review of "Cloud Detection over Snow and Ice with Oxygen A- and B-band Observations from the Earth Polychromatic Imaging Camera (EPIC)"

_Atmospheric Measurement Techniques, 2019_

## Referee Comment (RC1) · Anonymous Referee #1 · 29 Oct 2019

This study presents an updated DSCOVR/EPIC cloud detection over snow and ice surfaces. It improves the current scheme by better accounting for changes in surface altitude and the solar or viewing zenith angles.

The topic is appropriate, the method physically sound, the general structure sensible, and the improvements in EPIC's cloud flagging look good. However, some of the radiative transfer choices seem unphysical, details needed for replication are missing, and discussion of several important issues is absent. Unfortunately, the revisions I propose may mean re-running the radiative transfer and recalculating the thresholds so I request major revisions.

[Figure]

I expect that the main conclusions of the paper to be solid and that the authors should have little trouble in dealing with my comments. With revisions I would judge the science and presentation to be of higher quality and would support its publication.

**1. Specific comments:**

**1.1 General flow and clarity** The order is sensible but important details sometimes appear late in the paper in a way that confused me. For example, I don't see an explicit statement that the training & validation is versus the GEO/LEO dataset until P11. This should be in the introduction and mentioned when talking about performance (e.g. P4L18). It's also not immediately clear what is new. So the old algorithm doesn't account for surface height (P16L11, 16 pages in!), but what else exactly? Please explain in the introduction, and see the line-by-line technical comments.

**1.2 Incomplete information regarding methodology** How did you get the regression statistics (e.g. P9L26–30)? I first assumed simultaneous multi-variate least squares, but P12L7 makes me think not. In Table 1 do you have error bars? I also think your equation is complex (see section below), how do you handle the imaginary part?

I don't see your snow & ice surface definition until P11L23–25 which I think says you're using GEO/LEO data, and only "permanent" snow, i.e. not seasonal? But then why are there so many samples over N America and Eurasia in Figure 7(a) but not in Figure 8(a)? If you develop using permanent snow then this needs to be said in the introduction and potential issues with e.g. snow-covered forests with lower albedo need to be discussed.

On P12L16–24 "when applied to a different dataset". What is this different dataset? Did you subsample the full dataset? Does this different set have the same distribution of SZA, time etc?

**1.3 Mathematical issues** P6L25 Eq. (7) and (8). Could you expand on the switch to $c_0$? I work it out as complex:

$$\ln\left(R_{abs}/R_{ref}\right) = mce^{-z/H}$$
$$-\ln\left(R_{abs}/R_{ref}\right) = -mce^{-z/H}$$
$$\ln\left(-ln(R_{abs}/R_{ref})\right) = \ln(-mce^{-z/H}) = \ln(-1) + \ln(mce^{-z/H})$$
$$\ln(-\ln R_{abs}/R_{ref})) = i\pi + \ln\left(c\right) + \ln\left(m\right) - z/H$$
$$c_0 = i\pi + \ln\left(K_a w_1 \rho_0 H\right)$$

Please explain my error or comment on how this affects your regression.

Minor points: dln(x) is widely used in calculus, how about something without another standard meaning, like dbln(x) or ln"(x). I also think you lost an $(R_{abs}/R_{ref})$ in Eq.(9) on P7L5.

**1.4 Radiative transfer (RT) description & choices**

I think that the general approach is sensible but some details aren't clear and several RT inputs are physically unrealistic. These are my biggest technical issue with the paper and are the primary reason I propose major revisions.

You simulate liquid clouds above 2.5 km, consistently 1 km thick, and over frozen surfaces up to 15 km in altitude with albedo of 0.8.

Firstly, CALIOP sees liquid in Arctic clouds <2.5 km (Cesana et al. 2012, doi: 10.1029/2012GL053385) and the ARM site in Alaska also sees lots of these lower clouds (e.g. Zhao & Wang 2010, doi: 10.1029/2010JD014285). Your higher clouds should typically be ice, which may affect both $R_{ref}(\tau)$ and the in-cloud path lengths.

I would like to see sub-2.5 km clouds included in your RT. You might need to exclude them from your threshold calculation to prevent too many false positives, but these low clouds are particularly difficult for LEO/GEO-based infrared detection. The implications of this for your testing & validation should be discussed and even if you find you can't reliably test for these low clouds, then you should be explicit about this limitation.

The fixed geometric thickness also might affect your thresholds somewhat. A 1 km thick liquid cloud with $\tau = 3$ is very low $N_d$ and should have unrealistically large within-cloud

path lengths. This might contribute to the discussion on P10L15–19. I'd propose a thickness that varies realistically with $\tau$ based on number concentration or a published relationship (e.g. for liquid clouds Eq. 2 from Chiu et al. 2014, doi: 10.5194/acp-14-8389-2014).

Your 0.8 albedo for both bands needs support. At the very least, you need to consider what this means for e.g. snow covered forests where the albedo is substantially lower, and may vary between the bands. Perhaps some simple physical argument with discussion of the limitations might be enough.

Finally, I don't know surfaces on Earth >10 km altitude. Why include your 15 km surfaces? Your observation sample should lack such cases, does this affect the regression statistics for the RT sample in Table 1?

I'd also appreciate some other details. The paragraph P8L9–16 is a good place to explicitly state that within-cloud absorption by $O_2$ is included in your RT (it is, right?). I also assume the EPIC ILS are broad enough that line broadening barely matters but would like a comment on this plus a reference to your spectroscopic database.

**Summary**: I believe your RT should include clouds <2.5 km (which may be liquid, supported by appropriate references), but higher clouds should contain ice and geometric thicknesses should vary realistically in tau. Then just recalculate the thresholds and statistics. I suspect your results will be robust to these choices, but some details could differ and your results would be more physically defensible. I think you should also at least discuss the very high surface altitudes and whether your regressions are affected.

**1.5 Discussion of LEO/GEO limitations**

It's fair enough that you test versus GEO/LEO, but you should explain how their limitations are relevant to your analysis. Examples of the sorts of references that should be included in the discussion are Wang et al. (2016, doi: 10.1002/2016JD025239) for MODIS collection 6, Karlsson & Håkansson (2018, doi: 10.5194/amt-11-633-2018) for

AVHRR and Shang et al. (2018, doi: 10.1038/s41598-018-19431-w) for Himawari-8.

**Technical comments**

**2.1 General:** Please check for missing articles or pluralisation where you currently treat normal nouns as proper nouns. Example insertions in square brackets:

P4L2 "the height of [the] effective reflective layer"
P4L4: "for use over [the] land surface" or "for use over land surface[s]"
P4L12: "but [a] large discrepancy is found"
P4L17 "over snow/ice surface" → "over snow- or ice-covered surfaces" (the hyphens here are an opitional style choice)

There are others. Plus the Andes, which takes the definite article despite its capitalisation:

P15L30: "…and the southern tip of [the] Andes"
P16L12–13: "…southern tip of [the] Andes could…"

**2.2 Line-by-line**

P1L24–26: When talking about performance statistics, please mention against what you are comparing. Some form of: "against a product based on multiple other passive sensors" or similar.

P3L1–7 para: "reference channel", change to "weakly absorbing reference channel" to help those unfamiliar with the approach.

P1L27: "Less significant" – I don't see significance tests, perhaps "less substantial"?

P2L7–8: not sure what long haul means.

P2L12: "narrow" is not an absolute. On P3L3 you mention the channel centres, could you add typical FWHM or another statistic that describes the spectral width there?

P2L19: "winter", "summer" please specify (I assume) "boreal"

P4L10: "are performing reasonably well" – this value judgment depends on assumptions about the performance of other cloud flags. I would prefer "show good agreement" with some performance statistic(s) in brackets.

P4L11–12: comment that "accuracy rate" and "correct cloud detection rate" will be defined later, or describe here.

P5L2: "based on well-known and well-mixed atmospheric $O_2$ gaseous absorption" – this looks to me like the adjectives both refer to "absorption", but it isn't exactly "well-mixed absorption" you mean. How about something like "well-known gaseous absorption of well-mixed atmospheric $O_2$".

P5L3: "..gaseous absorption, therefore, changes in observed radiance in the expected $O_2$ band" – I find "therefore" a weird link here, I don't think it's the $O_2$ band that's "expected". How about "...gaseous absorption. Changes in observed radiance in the $O_2$ band are expected to contain..."

P8L14 "convoluted with" → "convolved with" (I believe this is the verb for mathematical convolution, please check).

P8L21–22: a reference or pointer to the atmosphere definitions would be handy.

P9L1: "duplicate the quantitative relationship"... maybe "simplified relationship"? The RT model uses quantitative relationships too.

P14L23–28: this is a stylistic preference, but why pick a, b, c, d? In my opinion the standard notation (TP/TN/FP/FN for True/False Positive/Negative) is more easily understood and would make help me to interpret Equations (11)–(13) on sight.

P16L2: "Comparison show that..." → "Comparison shows that" (typo missing "s")

P16L7–8 – "indicates high cloud fraction (>80 %) over...", I'd just say "indicates cloud fraction > 80 % over..." because "high cloud fraction (>80 %)" could also be >80 % coverage of high-altitude clouds. This also appears on P14L17, where "high cloud fraction"

is >95 %. I'd be tempted to change "low cloud fraction (<5 %) and high cloud fractions (>95 %) categories" to "cloud fraction < 5 % and cloud fraction > 95 % categories".

P16L17: "achieved high accuracy. . ." → "has improved accuracy" ("high" again seems too subjective to me).

P17L1–3 this explanation seems physical but I don't think it's accessible to a non-specialist. How about: "This method is based on the fact that photons reflected by clouds above the surface will travel, on average, a shorter distance through the atmosphere and so experience less absorption by $O_2$" or similar?

P17L16: "these performance matrices". I would prefer "metrics" because you haven't explicitly introduced results as a matrix previously, and also these values are not the matrix itself, but derived from it (e.g. accuracy score = trace of normalised confusion matrix).

P17L11: "Model derived algorithm is chosen because of its stable performance". Do you mean that you chose the model algorithm because it performs better for the sample that was not used in training the obs-based dataset? If so, please change sentence to say this and, as requested earlier, describe how the datasets differ.

P26L9: ". . .on the right side of black lines will be identified as clear sky. . .": this implies that you use the black lines as a threshold, but I think you prefer the red dashed lines. Please rephrase to be clear that the black lines are a possible selection but you don't use them (if this is true).

P27L5: This is very nitpicky, but the (d) colour bar makes it look like you have continuous cloud mask values. I'd personally change the colour bar tick mark locations to be the actual flag values (1, 2, 3, 4 instead of 1.0, 1.6,. . .)

P28L1 : Figure 5, could you add a legend or some text indicator on one of the panels for the colours? This isn't vital given it's in the caption, but it would be nicer.

P32L5: Figure 9 caption: "matrix" → "metrics" as above.

---

## Referee Comment (RC2) · Anonymous Referee #2 · 22 Nov 2019

Zhou et al. described a cloud detection algorithm over snow and ice with oxygen A and B band. They have demonstrated that the new cloud mask algorithm is an improvement compared to the current EPIC cloud mask algorithm. They derived an analytic relationship between the double logarithm of the O2 band ratios and the surface elevation and the zenith angles. They also showed the limit of the algorithm for optically thin clouds and low elevations. I think the paper is fit the topic of AMT.

Specific comments

Page 8 lines 9-16. The authors described briefly the radiative transfer simulator for EPIC. Does the simulator have sphericity correction at the solar and viewing zenith

angles larger than 80 degree?

Page 8 lines 25-28.

Why the surface height in the simulations is from 0 to 15 km with 2.5 km increment? The surface height larger 9 km is not useful and the increment of 2.5 km is too large. The increment of 0.5 km would be a better option.

Page 9, lines 26 – 30 It is not clear how the coefficients were derived. Could you explain it in detail?

Page 10 lines 24-26 How did you select the snow/ice surfaces?

Page 12 lines 4-9 Please explain more details about the regression. How did you design the model to predict the median . . .?

Page 12 lines 20 -21 What 'non-negligible uncertainties' do you mean here?

Fig. 1 I,j

The 'Fitted threshold' is not easy to understand. Do you mean the fitted A-band and B-band ratio? Do you use the simulated A-band ratio, m, z, to derive the coefficients in Table1, then calculate the 'Fitted threshold' using these coefficients? Fig.1i,j shows that the fit is almost linear. Will it cause scatter if the coefficients are applied to other data not in the simulations? If the surface albedo is 0.6 or 0.9, could you get the same coefficients?

Fig.2 Since the algorithm also detects clouds over snow/ice on top of mountains, could you make a similar plot for surface height of 2.5 km or 5 km?

Fig. 3 How do you explain the scatter in the clear-sky plots?

Fig. 4 It seems that you have to use more digits in the colorbar for (a,b). For (d) please use integer in the colorbar.

---

## Author Comment (AC1) · 27 Jan 2020

This study presents an updated DSCOVR/EPIC cloud detection over snow and ice surfaces. It improves the current scheme by better accounting for changes in surface altitude and the solar or viewing zenith angles.

The topic is appropriate, the method physically sound, the general structure sensible, and the improvements in EPIC's cloud flagging look good. However, some of the radiative transfer choices seem unphysical, details needed for replication are missing, and discussion of several important issues is absent. Unfortunately, the revisions I pro- pose may mean re-running the radiative transfer and recalculating the thresholds so I request major revisions.

I expect that the main conclusions of the paper to be solid and that the authors should have little trouble in dealing with my comments. With revisions I would judge the science and presentation to be of higher quality and would support its publication.

Thank you for a thorough review of the paper and many insightful comments.

**1. Specific comments:**

**1.1 General flow and clarity** The order is sensible but important details sometimes appear late in the paper in a way that confused me. For example, I don't see an explicit statement that the training & validation is versus the GEO/LEO dataset until P11. This should be in the introduction and mentioned when talking about performance (e.g. P4L18). It's also not immediately clear what is new. So the old algorithm doesn't account for surface height (P16L11, 16 pages in!), but what else exactly? Please explain in the introduction, and see the line-by-line technical comments.

Thank you for pointing out this. We introduced the validation dataset in the introduction and also explicitly stated that limitation of current algorithm by using fixed threshold.

**1.2 Incomplete information regarding methodology** How did you get the regression statistics (e.g. P9L26–30)? I first assumed simultaneous multi-variate least squares, but P12L7 makes me think not. In Table 1 do you have error bars? I also think your equation is complex (see section below), how do you handle the imaginary part?

The threshold values are indeed derived using multivariate least squares regression, but only from the clear sky simulations (Figure 1, i-j) and clear sky observations (Figure 5, a-b). We have mentioned the particular methodology in multiple places (P7L20-25, P11L1-5, P14L5-7) now.

We added multiple correlation coefficients in Table 2.

In the original derivation, we missed a negative sign in Eq. (6), thus after multiplying (-1) after first logarithmic function, there is no negative sign in Eq. (8).

I don't see your snow & ice surface definition until P11L23–25 which I think says you're using GEO/LEO data, and only "permanent" snow, i.e. not seasonal? But then why are there so many samples over N America and Eurasia in Figure 7(a) but not in Figure 8(a)? If you develop using permanent snow then this needs to be said in the introduction and potential issues with e.g. snow-covered forests with lower albedo need to be discussed.

Thanks for pointing this out. We actually included seasonal snow and ice over water categories in selecting the collocation dataset. The snow/ice cover information was included in the Langley GEO/LEO composite dataset, which was based on the Near-real-time Ice and Snow Extent (NISE) data set from the National Snow & Ice Data Center (NSIDC). We added more details to the text.

We have stated in the introduction that current work is focused on cloud mask over snow/ice surfaces. We've also added radiative transfer simulations for surface albedo at 0.6 and 1.0 to cover the range of snow and ice albedos. Results show that within the solar zenith angle range where EPIC does its retrieval, the clear sky A-band and B-band ratios are not sensitive to surface albedo.

On P12L16–24 "when applied to a different dataset". What is this different dataset? Did you subsample the full dataset? Does this different set have the same distribution of SZA, time etc?

We derived the set of regression coefficients using a training dataset from January and July 2017. A different dataset here refers to similar data but from different months, e.g., January and July of 2016. We changed the "different dataset" to "different data period".

**1.3 Mathematical issues** P6L25 Eq. (7) and (8). Could you expand on the switch to $c_0$? I work it out as complex:

$$\ln(R_{abs}/R_{ref}) = mce^{-z/H}$$
$$-\ln(R_{abs}/R_{ref}) = -mce^{-z/H}$$
$$\ln(-\ln(R_{abs}/R_{ref})) = \ln(-mce^{-z/H}) = \ln(-1) + \ln(mce^{-z/H})$$
$$\ln(-\ln R_{abs}/R_{ref})) = i\pi + \ln(c) + \ln(m) - z/H$$
$$c_0 = i\pi + \ln(K_a w_1 \rho_0 H)$$

Please explain my error or comment on how this affects your regression.

As mentioned above, we missed a negative sign in Eq. 6, thus there should be no negative sign in Eq. 8.

Minor points: dln(x) is widely used in calculus, how about something without another standard meaning, like dbln(x) or ln''(x). I also think you lost an ($R_{abs}/R_{ref}$) in Eq.(9) on P7L5.

Thanks for the suggestion. We used dbln throughout in the text and added missing ($R_{abs}/R_{ref}$).

**1.4 Radiative transfer (RT) description & choices**

I think that the general approach is sensible but some details aren't clear and several RT inputs are physically unrealistic. These are my biggest technical issue with the paper and are the primary reason I propose major revisions.

You simulate liquid clouds above 2.5 km, consistently 1 km thick, and over frozen surfaces up to 15 km in altitude with albedo of 0.8.

Firstly, CALIOP sees liquid in Arctic clouds <2.5 km (Cesana et al. 2012, doi: 10.1029/2012GL053385) and the ARM site in Alaska also sees lots of these lower clouds (e.g. Zhao & Wang 2010, doi: 10.1029/2010JD014285). Your higher clouds should typically be ice, which may affect both $R_{ref}(\tau)$ and the in-cloud path lengths.

We took the effort to implement ice cloud in the RT model. The analysis now uses ice cloud simulations instead of water cloud.

I would like to see sub-2.5 km clouds included in your RT. You might need to exclude them from your threshold calculation to prevent too many false positives, but these low clouds are particularly difficult for LEO/GEO-based infrared detection. The implications of this for your testing & validation should be discussed and even if you find you can't reliably test for these low clouds, then you should be explicit about this limitation.

Instead of using even increment of cloud top height starting from 2.5km, we now use cloud top height from 1, 3, 5 km, then increase 2.5 km afterwards. The figures (Figure 3a and 3b) show results for cloud top at 1, 3, 5 km, because for higher clouds, the band ratios have large sensitivity; hence they are not of main concern.

The fixed geometric thickness also might affect your thresholds somewhat. A 1 km thick liquid cloud with $\tau = 3$ is very low $N_d$ and should have unrealistically large within-cloud path lengths. This might contribute to the discussion on P10L15–19. I'd propose a thickness that varies realistically with $\tau$ based on number concentration or a published relationship (e.g. for liquid clouds Eq. 2 from Chiu et al. 2014, doi: 10.5194/acp-14- 8389-2014).

The Chiu et al. 2014 study was based on data from ARM SGP site and may not be applicable directly to the polar regions. The CALIOP data shows quite a large range of geometric thickness and optical thickness in Antarctic clouds. To test cloud geometrical thickness sensitivity, we

conducted additional radiative transfer simulations with cloud thickness varying from 0.5 km to 4km. Results from these sensitivities are added in figure 4.

Your 0.8 albedo for both bands needs support. At the very least, you need to consider what this means for e.g. snow covered forests where the albedo is substantially lower, and may vary between the bands. Perhaps some simple physical argument with discussion of the limitations might be enough.

We conducted additional clear sky and cloudy sky simulations with surface albedo of 0.6 and 1.0 to cover a broader range of potential snow and ice albedo. Our thresholds derivation only needs clear sky simulations. For which case, the oxygen band ratios vary very little for changes in surface albedo from 0.6 to 1.0 except when zenith angle is very large ($> 75°$). Thus thresholds derived with surface albedo 0.8 can be applied to all snow and ice surfaces with little problem. For cloudy sky simulations, as expected, the sensitivity of oxygen band ratios to clouds are higher for darker surfaces.

Finally, I don't know surfaces on Earth >10 km altitude. Why include your 15 km surfaces? Your observation sample should lack such cases, does this affect the regression statistics for the RT sample in Table 1?

Thanks for pointing this out. Our original thought was that the sensitivity to surface height can also provide information on the sensitivity to clouds (if we regard clouds as hard targets). We now limit the surface height to 7.5 km maximum. The regression coefficients are very similar. We updated the figures 6-11 using the new coefficients. The difference with the old version is very small.

In addition, we conducted cloud sensitivity with surface height of 2.5 km. As expected, results show that that higher surface elevation tends to make cloud detection more difficult.

I'd also appreciate some other details. The paragraph P8L9–16 is a good place to explicitly state that within-cloud absorption by O2 is included in your RT (it is, right?). I also assume the EPIC ILS are broad enough that line broadening barely matters but would like a comment on this plus a reference to your spectroscopic database.

**1: Yes, the O2 absorption within clouds is considered. This is done by assuming a fixed $O_2$ molecule vertical profile (US standard or other specified atmosphere).**
**2: line broadening caused by pressure and also line absorption parameters depending on temperature is considered. A high-resolution line by line calculation is first done in $O_2$ A- and B-band and then the results are convolved with the filter transmission function of EPIC. The line parameter database is HITRAN. ARTS (Atmospheric Radiative Transfer Simulator) is used to calculate the gas absorption cross section from the HITRAN line parameters. Additional information of RT model is added in the text.**

**Summary**: I believe your RT should include clouds <2.5 km (which may be liquid, supported by appropriate references), but higher clouds should contain ice and geometric thicknesses should vary realistically in tau. Then just recalculate the thresholds and statistics. I suspect your results

will be robust to these choices, but some details could differ and your results would be more physically defensible. I think you should also at least discuss the very high surface altitudes and whether your regressions are affected.

We conducted more RT sensitivity analysis following both reviewers' suggestions.

1) For clear skies, we included additional simulations for different surface albedo values (new Figure 2). We found that clear sky $O_2$ band ratios are not sensitive to surface albedo (in the 0.6~1.0 range) except for high zenith angles. Note cloud mask thresholds are derived with clear sky simulation data. We discarded surface elevation greater than 7.5 km cases.

2) For cloudy skies, since it's over cold regions (snow/ice surfaces) we used ice cloud in the simulation instead of water cloud. Besides variations in cloud optical thickness and cloud height (more low clouds), we tested the sensitivity due to surface albedo, cloud geometric thickness and surface elevation. The new Figure. 4 shows how cloud sensitivity changes with various parameters at the low zenith angles.

**1.5 Discussion of LEO/GEO limitations**

It's fair enough that you test versus GEO/LEO, but you should explain how their limitations are relevant to your analysis. Examples of the sorts of references that should be included in the discussion are Wang et al. (2016, doi: 10.1002/2016JD025239) for MODIS collection 6, Karlsson & Håkansson (2018, doi: 10.5194/amt-11-633-2018) for AVHRR and Shang et al. (2018, doi: 10.1038/s41598-018-19431-w) for Himawari-8.

Thank you for the suggestion and references. The GEO/LEO cloud detection is now discussed in the text in Section 4 when detailed GEO/LEO cloud data is introduced.

**Technical comments**
**2.1 General:** Please check for missing articles or pluralisation where you currently

treat normal nouns as proper nouns. Example insertions in square brackets:

P4L2 "the height of [the] effective reflective layer"
P4L4: "for use over [the] land surface" or "for use over land surface[s]"
P4L12: "but [a] large discrepancy is found"
P4L17 "over snow/ice surface" → "over snow- or ice-covered surfaces" (the hyphens here are an opitional style choice)

There are others. Plus the Andes, which takes the definite article despite its capitalisation:

P15L30: ". . .and the southern tip of [the] Andes" P16L12–13: ". . .southern tip of [the] Andes could. . ."

Done.

**2.2 Line-by-line**

P1L24–26: When talking about performance statistics, please mention against what you are comparing. Some form of: "against a product based on multiple other passive sensors" or similar.

Done.

P3L1–7 para: "reference channel", change to "weakly absorbing reference channel" to help those unfamiliar with the approach.

Done.

P1L27: "Less significant" – I don't see significance tests, perhaps "less substantial"? P2L7–8: not sure what long haul means.

Fixed.

P2L12: "narrow" is not an absolute. On P3L3 you mention the channel centres, could you add typical FWHM or another statistic that describes the spectral width there?

FWHMs are added in P3L5-6.

P2L19: "winter", "summer" please specify (I assume) "boreal"

Done.

P4L10: "are performing reasonably well" – this value judgment depends on assumptions about the performance of other cloud flags. I would prefer "show good agreement" with some performance statistic(s) in brackets.

Overall CM accuracy of 80.2% and 85.7% correct cloud detection rate are described in the next sentence.

P4L11–12: comment that "accuracy rate" and "correct cloud detection rate" will be defined later, or describe here.

Add "accuracy and correct cloud detection rate are defined in Section 5".

P5L2: "based on well-known and well-mixed atmospheric $O_2$ gaseous absorption" – this looks to me like the adjectives both refer to "absorption", but it isn't exactly "well- mixed absorption" you mean. How about something like "well-known gaseous absorption of well-mixed atmospheric $O_2$".

Suggestion followed. Thanks!

P5L3: "..gaseous absorption, therefore, changes in observed radiance in the expected $O_2$ band" – I find "therefore" a weird link here, I don't think it's the $O_2$ band that's "expected". How about

". . .gaseous absorption. Changes in observed radiance in the $O_2$ band are expected to contain. . ."

Thanks again for the suggestion.

P8L14 "convoluted with" → "convolved with" (I believe this is the verb for mathematical convolution, please check).

"Convoluted" changed to "convolved"

P8L21–22: a reference or pointer to the atmosphere definitions would be handy.

References to these atmospheric profiles are added.

P9L1: "duplicate the quantitative relationship". . . maybe "simplified relationship"? The RT model uses quantitative relationships too.

"quantitative" changed to "simplified".

P14L23–28: this is a stylistic preference, but why pick a, b, c, d? In my opinion the standard notation (TP/TN/FP/FN for True/False Positive/Negative) is more easily understood and would make help me to interpret Equations (11)–(13) on sight.

Suggestions taken.

P16L2: "Comparison show that. . ." → "Comparison shows that" (typo missing "s")

Fixed.

P16L7–8 – "indicates high cloud fraction (>80 %) over. . .", I'd just say "indicates cloud fraction > 80 % over. . ." because "high cloud fraction (>80 %)" could also be >80 % coverage of high-altitude clouds. This also appears on P14L17, where "high cloud fraction" is >95 %. I'd be tempted to change "low cloud fraction (<5 %) and high cloud fractions (>95 %) categories" to "cloud fraction < 5 % and cloud fraction > 95 % categories".

Suggestion taken.

P16L17: "achieved high accuracy. . ." → "has improved accuracy" ("high" again seems too subjective to me).

Done

P17L1–3 this explanation seems physical but I don't think it's accessible to a non- specialist. How about: "This method is based on the fact that photons reflected by clouds above the surface will travel, on average, a shorter distance through the atmo- sphere and so experience less absorption by $O_2$" or similar?

Suggestion taken. Thanks!

P17L16: "these performance matrices". I would prefer "metrics" because you haven't explicitly introduced results as a matrix previously, and also these values are not the matrix itself, but derived from it (e.g. accuracy score = trace of normalised confusion matrix).

Thanks for pointing this out. It was meant to be "metrics".

P17L11: "Model derived algorithm is chosen because of its stable performance". Do you mean that you chose the model algorithm because it performs better for the sample that was not used in training the obs based dataset? If so, please change sentence to say this and, as requested earlier, describe how the datasets differ.

Done.

P26L9: ". . .on the right side of black lines will be identified as clear sky. . .": this implies that you use the black lines as a threshold, but I think you prefer the red dashed lines. Please rephrase to be clear that the black lines are a possible selection but you don't use them (if this is true).

The figure caption is modified for clarification.

P27L5: This is very nitpicky, but the (d) colour bar makes it look like you have continuous cloud mask values. I'd personally change the colour bar tick mark locations to be the actual flag values (1, 2, 3, 4 instead of 1.0, 1.6,. . .)

Color bar ticks are modified.

P28L1 : Figure 5, could you add a legend or some text indicator on one of the panels for the colours? This isn't vital given it's in the caption, but it would be nicer.

Bar legend is added.

P32L5: Figure 9 caption: "matrix" $\rightarrow$ "metrics" as above.

Fixed

---

## Author Comment (AC2) · 27 Jan 2020

Zhou et al. described a cloud detection algorithm over snow and ice with oxygen A and B band. They have demonstrated that the new cloud mask algorithm is an improvement compared to the current EPIC cloud mask algorithm. They derived an analytic relationship between the double logarithm of the O2 band ratios and the surface elevation and the zenith angles. They also showed the limit of the algorithm for optically thin clouds and low elevations. I think the paper is fit the topic of AMT.

Thank you for reviewing the paper and providing thoughtful comments.

Specific comments

Page 8 lines 9-16. The authors described briefly the radiative transfer simulator for EPIC. Does the simulator have sphericity correction at the solar and viewing zenith angles larger than 80 degree?

The sphericity is not considered in the model. This is not a problem for EPIC, as the standard Level 2 cloud products are only generated for view zenith angle < $76^0$

Page 8 lines 25-28.

Why the surface height in the simulations is from 0 to 15 km with 2.5 km increment? The surface height larger 9 km is not useful and the increment of 2.5 km is too large. The increment of 0.5 km would be a better option.

We removed simulations beyond surface height above 7.5km. It would be better to use increment of 0.5 km in height, but since the function with height is close to linear, we didn't redo the clear sky simulation.

Page 9, lines 26 – 30 It is not clear how the coefficients were derived. Could you explain it in detail?

We used a multivariate linear regression to do the fitting. The regression takes surface elevation (Z) and ln (m) as two independent variables and db ln ($R_{abs}/R_{ref}$) as dependent variable. The derived coefficients are used to prediction expected db ln ($R_{abs}/R_{ref}$) and then $R_{abs}/R_{ref}$ . More details are added in the text.

Page 10 lines 24-26 How did you select the snow/ice surfaces?

Initially we used surface albedo of 0.8 to represent snow and ice surfaces in the model. Additional simulations are performed for surface albedo at 0.6 and 1.0 to cover the range of albedos over snow, sea ice surfaces. In the observational data, we used the surface type information included in the Langley GEO/LEO composite data, which is based on the IGBP surface type dataset and the Near-real-time Ice and Snow Extent (NISE) data set from the National Snow & Ice Data Center (NSIDC). We have added detailed references in the text.

Page 12 lines 4-9 Please explain more details about the regression. How did you design the model to predict the median …?

The same multivariate linear regression is applied to the observational data. The nature of regression is to provide a function that minimizes the total squared error which will approximately pass cross the median of each sample bins. But our cloud mask threshold is to find the upper bound of clear sky value so that all clear sky pixels will be under that curve idealistically. In reality, because there are many overlaps between clear and cloudy pixels as shown in Figure 5c and 5d, we could only move the divider up slightly to balance the clear and cloudy detection.

Page 12 lines 20 -21 What 'non-negligible uncertainties' do you mean here? Fig. 1 I,j

The reference cloud mask we used is based on GEO/LEO retrievals, which has its own uncertainties. Cloud contamination is one of the main causes of scatter in the clear sky regression. Other causes may include uncertainties in geolocation, surface elevation, atmospheric profile etc..

The 'Fitted threshold' is not easy to understand. Do you mean the fitted A-band and B-band ratio? Do you use the simulated A-band ratio, m, z, to derive the coefficients in Table1, then calculate the 'Fitted threshold' using these coefficients? Fig.1i,j shows that the fit is almost linear. Will it cause scatter if the coefficients are applied to other data not in the simulations? If the surface albedo is 0.6 or 0.9, could you get the same coefficients?

You are right. The fitted threshold refers to A-band and B-band ratios computed with regression coefficients. Ideally, everything being equal, the ratios for cloud sky should be larger than that of a clear sky. As mentioned earlier, we are trying to find the upper bound of the clear sky ratios. The regression is derived with simulations using surface albedo of 0.8. To test if these coefficients work for other surface albedos, we conducted new clear sky sensitivities with surface albedo of 0.6 and 1.0 and results are shown in Fig. 2. In majority of the cases, the clear sky A-band and B-band ratios are not sensitive to surface albedo, the fitting is problematic at large zenith angles (>76°) that EPIC does not retrieve.

Fig.2 Since the algorithm also detects clouds over snow/ice on top of mountains, could you make a similar plot for surface height of 2.5 km or 5 km?

We added cloudy sky sensitivities for surface elevation of 2.5 km (Figure 4). Compared to surface at sea level, the cloud detection algorithm is less sensitive over high mountains; more thin and low clouds will be undetected. Discussions are added to the text.

Fig. 3 How do you explain the scatter in the clear-sky plots?

The reference cloud mask used here is based on multi-sensor, including those from both geosynchronous orbit (GEO) or low Earth orbits (LEO). In general, the sensors used have better cloud detection capabilities than EPIC, but misclassifications still exist. The scatter we see can come from multiple sources, include cloud contamination, surface elevation uncertainty, cross-sensor consistency, geolocation error, atmospheric profile uncertainties, etc.

Fig. 4 It seems that you have to use more digits in the colorbar for (a,b). For (d) please use integer in the colorbar.

Done.

---

## Author Response (AR2)

I commend the authors for a thorough and professional response, and now that I can follow the data choices and logic I see how this is a sensible approach to improve atmospheric measurement. I fully support publication but have a bunch of minor comments below. Mostly they're very small changes to improve readability and clarify logic, but I have one slightly larger comment about Figure 7.

Thank you again for the kind comments.

Figure 7 is fine as it is, but could be more informative as a 4x4 annotated heatmap or confusion matrix-like figure. The 4x4 elements would contain the % of each sample for each pairwise condition for (at least) LEO/GEO vs v01 and LEO/GEO vs new and (optionally) v01 versus new. The diagonal elements would be agreement between categories, and the off-diagonals the disagreement (e.g. LEO/GEO cloud fraction <5 % but cloud flag > 1). This would contain the same information but be a lot easier for readers to get figures from. I leave this up to the authors though, but if you want to keep the current version then please match the scales on the y axes in each row (see a vs b, c vs d), to allow immediate visual comparison.

We changed Figure 7 according to reviewer's suggestion.

MINOR COMMENTS:
p2L13: "UV and Vis/NIR" - suggest spelling out on first use

Done p2L26: Replace "fewer spectral channels available compared..." with "fewer spectral channels compared...".

Done.

p2L28: typo, need to remove "s" from "spectrometers" in the MODIS name.

Done.

p3l21: typo, "cloud edge[s]" (insert "s" since we're talking multiple).

Done.

p4L7: flipped words: "...the otherwise same..." --> "otherwise the same"

Done.

p11L4: "...as suggested in Eq. (9)". I don't believe the equation is making the suggestion, but you are making the suggestion and then applying Eq. (9)? Suggested text: "...for the simulations, with the sample restricted to a zenith angle difference of below 6$^\circ$".

Eq. (9) does suggestion a linear regression. But sample restriction is not suggested by the equation. Text is modified.

p13L4: General comment for the section, could you add information on sample size? It would be more pleasant to know that without having to squint at figure labels and try to work it out in my head.

Sample size for each month is added in Figure 7.

p14L19: "median" replace with "mean"? I'm nitpicking again, but OLS is more commonly introduced as mean unbiased estimator. Sure, if your Gaussian assumptions work then mean=median, but by mentioning median I start to think of other estimators, particularly non-parametric approaches to use medians.

"Median" is changed to "mean"

p14L23-26: How did you define "best"? Maximise accuracy score, or what? Please specify.

Here we check all three rates, the accuracy, correct detection rate and false detection rate. The accuracy rate generally reflects both correct detection rate and false detection rate, but in case the samples are not evenly distributed, we would sacrifice the accuracy rate a little bit to guarantee the false detection rate is not too high or correct detection rate is not too low. We added some description in the text.

p15L24-26: do LEO/GEO errors vary as you look at different parts of Antarctica? If you have a comment and/or reference here it would be nice. This is not vital though.

Wang et al. 2016 show that misidentification of clear as cloud also occurs quite frequently in Eastern Antarctica during boreal spring and fall.

p19L15: I would delete "the summer month" and just say "in July". Avoids the boreal vs austral summer issue.

Done.

p19L25-26: "Therefore the method presented in this work provides a solution to polar cloud detection when infrared channels are not available"... I don't like the logic of this sentence. Your prior comment is that this approach is complementary to, and improves upon, infrared. So you don't have this *only* because of a lack of infrared channels. I'd change the last part to "...when infrared channels are not available, or struggle to distinguish between cloudy and clear scenes."

Changed according to the suggestion. Thank you!

Figure 1c,d: the axis label still says dln, please change to dbln

Fixed.

General language comments:
There are a few grammatical article issues still. Mostly noted when standard nouns have adjectives on them, e.g. when "clouds" or "surfaces" are described. Changing snow/ice surface to "snow and ice surfaces" would fix a lot of these.

Other examples, I'll put <> to represent suggested deletion and [] for insertion:
p2L29: "<the> water and ice clouds" ("the" not necessary since "clouds" is plural)
p3L31: "...and [the] sea surface" (singular non-proper noun)
p4L1: "..at <the> 680 nm and 780 nm" (using 680 nm as a proper noun standing in for "the 680 nm channel")
p19L10: "[The] model derived algorithm" ("algorithm" singular)
p19L12: "...accuracy of [the] EPIC cloud mask" ("cloud mask" singular object)

The manuscript was proofread again.

[revised manuscript text omitted]